# Failure-Aware Gaussian Process Optimization with Regret Bounds

**Shogo Iwazaki**
MI-6 Ltd., Tokyo, Japan
iwazaki@mi-6.co.jp

**Shion Takeno**
RIKEN AIP, Tokyo, Japan
shion.takeno@riken.jp

**Tomohiko Tanabe**
MI-6 Ltd., Tokyo, Japan
tanabe@mi-6.co.jp

**Mitsuru Irie**
MI-6 Ltd., Tokyo, Japan
irie@mi-6.co.jp

## Abstract

Real-world optimization problems often require black-box optimization with observation failure, where we can obtain the objective function value if we succeed, otherwise, we can only obtain a fact of failure. Moreover, this failure region can be complex by several latent constraints, whose number is also unknown. For this problem, we propose a failure-aware Gaussian process upper confidence bound (F-GP-UCB), which only requires a mild assumption for the observation failure that an optimal solution lies on an interior of a feasible region. Furthermore, we show that the number of successful observations grows linearly, by which we provide the first regret upper bounds and the convergence of F-GP-UCB. We demonstrate the effectiveness of F-GP-UCB in several benchmark functions, including the simulation function motivated by material synthesis experiments.

## 1   Introduction

The effectiveness of optimization methods based on Gaussian process (GP) regression for expensive-to-evaluate black-box functions has been repeatedly shown in a wide range of real-world applications, including robotics [34, 35], experimental design [14], and hyperparameter optimization [4, 41]. On the other hand, failure of the observation itself must often be considered. For example, in the optimization of hyperparameters for a complicated physical model, the evaluation may crash for some hyperparameters. Another example is materials development, in which experimental testing of new materials can reveal that the synthesis procedure fails.

Therefore, this study considers the optimization of a black-box function $f$ with a black-box deterministic failure function $c : \mathcal{X} \rightarrow \{0, 1\}$, where $\mathcal{X}$ is an input space. In this study, failure of observation at the input $\boldsymbol{x}$ is represented as $c(\boldsymbol{x}) = 1$, while the success of observation is represented as $c(\boldsymbol{x}) = 0$. Then, our optimization problem can be formulated as follows:

$$\boldsymbol{x}^* = \arg\max_{\boldsymbol{x} \in \mathcal{X}} f(\boldsymbol{x}) \;\; s.t. \;\; c(\boldsymbol{x}) = 0, \tag{1}$$

where the observation of $f$ can be obtained only if $c(\boldsymbol{x}) = 0$, but the observation cost is consumed even if $c(\boldsymbol{x}) = 1$. Hence, the goal is to efficiently identify the optimal point $\boldsymbol{x}^*$ while considering the observation failure.

One of the notable technical difficulties of problem (1) is how to handle the failure function $c$. To model $c$, existing studies [32, 2] use the GP classification (GPC) model [38] or its variants, which assume that $c$ can be represented by a smooth latent function. However, the observation failure can

37th Conference on Neural Information Processing Systems (NeurIPS 2023).

be caused by several latent constraints implicitly. In addition, since only the binary failure can be obtained, the number of latent constraints is also unknown. Therefore, modeling $c$ with the usual GPC is hard and often unsuitable, which can degrade the optimization performance. Hence, an optimization method under a mild assumption on $c$ is demanded. Furthermore, the existence of $c$ makes theoretical analysis and even securing the number of successful observations difficult.

**Our contribution.** In this paper, we propose a novel GP-based optimization algorithm, failure-aware GP upper confidence bound (F-GP-UCB), which chooses the next evaluation in an input domain except for the adaptively adjusted neighborhood of the past observation failures. Furthermore, we show that the number of successful observations grows linearly under a very mild assumption of the failure function $c$ that the optimal solution lies on an interior of a feasible region. Then, we provide the first regret upper bound of the GP optimization problem (1), which shows that F-GP-UCB converges to the optimal solution with high probability. We demonstrate the effectiveness of the F-GP-UCB algorithm with several benchmark functions, including a heuristic simulation function motivated by materials research of quasicrystals.

**Related work.** In the past few decades, many GP-based optimization algorithms have been developed [37, 42, 18, 8, 53]. In addition to the standard optimization settings, various extensions have been studied, such as parallel [9], high-dimensional [54], multi-fidelity [25], and robust optimization [6].

GP-based constrained optimization [10, 11, 19, 47] has a close relation to our study. It considers the black-box optimization under an inequality constraint $g(\boldsymbol{x}) \leq 0$, where $g$ is also a black-box function. If the failure function in this paper is recast as $c(\boldsymbol{x}) = \mathbb{1}\{g(\boldsymbol{x}) > 0\}$, the optimization problem matches that of problem (1). However, there are two crucial differences. First, in constrained optimization, observations at an input can be obtained even if the constraint is not satisfied, i.e., $g(\boldsymbol{x}) > 0$. Under the setting of this paper, no information about the objective function can be obtained for input points that result in failure. Second, in constrained optimization problems, it is generally assumed that noisy observation of $g(\boldsymbol{x})$ is possible. In the setting of our study, we cannot obtain a direct observation from the latent function $g$.

There exist works on GP optimization that take into account failures. Lindberg and Lee [32] proposed an algorithm that combines the Expected Improvement (EI) criterion [37] with the posterior success probability of the classical GPC. Instead of GPC, Sacher et al. [39] used a support vector machine-based model. However, although $c$ is deterministic, these classifiers assume that observation failure is essentially stochastic, i.e., the evaluations at the same input can fail and succeed. This model misspecification can degrade the optimization performance, as described in Bachoc et al. [2]. Then, Bachoc et al. [2] proposed the deterministic variant of GPC which models $c$ as $c(\boldsymbol{x}) = \mathbb{1}\{g(\boldsymbol{x}) \geq 0\}$, where $g$ is a latent function modeled by a GP. They also provide an EI-based strategy and prove convergence to an optimal solution. From a practical point of view, surrogate models of $c$ navigate the optimization process efficiently when $c$ is well represented through a smooth latent function $g$. However, practical applications often have too complex failure processes to model with one latent function (e.g., the failure function $c$ depends on several latent functions.) Furthermore, it is difficult to know that such a complex failure structure exists beforehand. Finally, from a theoretical perspective, Bachoc et al. [2] give the convergence guarantee, but its rate and the regret-based analysis are not provided.

## 2 Preliminaries

**Problem setup.** Let $f : \mathcal{X} \to \mathbb{R}$ and $c : \mathcal{X} \to \{0, 1\}$ be an unknown fixed objective function and an unknown failure function, respectively, where $\mathcal{X} := [0, 1]^d$ is the input space. At each time step $t$, the user makes a selection $\boldsymbol{x}_t \in \mathcal{X}$ and obtains the failure label $c(\boldsymbol{x}_t)$. If $c(\boldsymbol{x}_t) = 0$, the user proceeds to make a noisy observation $y_t = f(\boldsymbol{x}_t) + \epsilon_t$ where $\epsilon_t$ is a noise term which is conditionally $\sigma$-sub-Gaussian. Our noise model is a mild one; examples include an arbitrary distribution over $[-\sigma, \sigma]$ and a Gaussian noise with variance below $\sigma^2$. Note that, in the case of $c(\boldsymbol{x}_t) = 1$ (failure), the user obtains no further information.

The user's goal is to efficiently identify the optimal solution $\boldsymbol{x}^*$ over the unknown feasible region $S_c := \{\boldsymbol{x} \in \mathcal{X} \mid c(\boldsymbol{x}) = 0\}$. For convenience, we rephrase the problem (1) using $S_c$ as follows:

$$\boldsymbol{x}^* = \arg\max_{\boldsymbol{x} \in S_c} f(\boldsymbol{x}). \tag{2}$$

We assume that $S_c \neq \emptyset$ and that there exists a unique solution $\boldsymbol{x}^*$. Furthermore, we define the failure region $F_c$ as $F_c = \mathcal{X} \setminus S_c$.

**Regret.** We employ the *regret* to evaluate the algorithm's performance. The regret $r_t$ at step $t$ is defined as $r_t = f(\boldsymbol{x}^*) - f(\hat{\boldsymbol{x}}_t)$ if $\hat{\boldsymbol{x}}_t \in S_c$, otherwise $r_t = f(\boldsymbol{x}^*) - \min_{\boldsymbol{x} \in \mathcal{X}} f(\boldsymbol{x})$, where $\hat{\boldsymbol{x}}_t \in \mathcal{X}$ is the algorithm's estimated solution[1]. In our definition of $r_t$, the algorithm is supposed to incur the worst-case regret when $\hat{\boldsymbol{x}}_t$ is not in $S_c$. Similar definitions are also used in performance metrics of GP-based constrained optimization [19].

**Regularity assumptions for the objective function.** As a regularity assumption, we assume that $f$ is an element of the reproducing kernel Hilbert space (RKHS) $\mathcal{H}_k$, corresponding to a known positive-definite kernel $k : \mathcal{X} \times \mathcal{X} \to \mathbb{R}$, and has a bounded Hilbert norm $\|f\|_{\mathcal{H}_k} \leq B$. Furthermore, the kernel $k$ is assumed to be normalized, namely $\forall \boldsymbol{x} \in \mathcal{X}, k(\boldsymbol{x}, \boldsymbol{x}) \leq 1$. These are common assumptions in existing GP optimization literature [42, 51, 8, 48].

**Gaussian process modeling.** Our algorithm uses the modeling information of $f$ from GP [38], using only the successful observations. First, we assume zero-mean GP with the covariance function $k$ as the prior of $f$. Next, we model the generating process of the value $y_t$ queried at $\boldsymbol{x}_t$ as $y_t = f(\boldsymbol{x}_t) + \epsilon_t$, where $\epsilon_t \sim \mathcal{N}(0, \sigma^2)$. We note that the assumptions on the GP prior on $f$ and Gaussian noise, as stated above, are assumptions only for constructing the GP model, which can differ from the underlying true function $f$ and the noise. Let $\mathcal{I}_t^{(S)} := \{i \in \{1, \ldots, t\} \mid c(\boldsymbol{x}_i) = 0\}$ be the index set of successful observations, $\boldsymbol{X}_t^{(S)} := (\boldsymbol{x}_i)_{i \in \mathcal{I}_t^{(S)}}$ be the corresponding inputs, $\boldsymbol{y}_t^{(S)} := (y_i)_{i \in \mathcal{I}_t^{(S)}}$, be the corresponding outputs, and $n_t := |\mathcal{I}_t^{(S)}|$ be the number of successful observations. In addition, we also define the failure index set $\mathcal{I}_t^{(F)} := \{i \in \{1, \ldots, t\} \mid c(\boldsymbol{x}_i) = 1\}$. Given $\boldsymbol{X}_t^{(S)}, \boldsymbol{y}_t^{(S)}$, the posterior distribution of $f(\boldsymbol{x})$ becomes a normal distribution, whose posterior mean $\mu_t(\boldsymbol{x})$ and posterior variance $\sigma_t^2(\boldsymbol{x})$ are given as follows:

$$\mu_t(\boldsymbol{x}) = \boldsymbol{k}(\boldsymbol{x}, \boldsymbol{X}_t^{(S)})^\top \left( \boldsymbol{K}(\boldsymbol{X}_t^{(S)}) + \sigma^2 \boldsymbol{I}_{n_t} \right)^{-1} \boldsymbol{y}_t^{(S)}, \tag{3}$$

$$\sigma_t^2(\boldsymbol{x}) = k(\boldsymbol{x}, \boldsymbol{x}) - \boldsymbol{k}(\boldsymbol{x}, \boldsymbol{X}_t^{(S)})^\top \left( \boldsymbol{K}(\boldsymbol{X}_t^{(S)}) + \sigma^2 \boldsymbol{I}_{n_t} \right)^{-1} \boldsymbol{k}(\boldsymbol{x}, \boldsymbol{X}_t^{(S)}). \tag{4}$$

Here, $\boldsymbol{I}_{n_t}$ is a $n_t \times n_t$ identity matrix. Furthermore, denoting the index of the $j$-th least element of $\mathcal{I}_t^{(S)}$ as $i_j$, $\boldsymbol{k}(\boldsymbol{x}, \boldsymbol{X}_t^{(S)}) \in \mathbb{R}^{n_t}$ represents a vector whose $j$-th element is $k(\boldsymbol{x}, \boldsymbol{x}_{i_j})$. Similarly, $\boldsymbol{K}(\boldsymbol{X}_t^{(S)}) \in \mathbb{R}^{n_t \times n_t}$ represents the kernel matrix whose $jk$-th element is $k(\boldsymbol{x}_{i_j}, \boldsymbol{x}_{i_k})$.

Lastly, we define the maximum information gain [42, 49] as a quantity representing the GP complexity. The maximum information gain $\gamma_{t;A}$ from observing $t$ number of data points over the set $A \subset \mathbb{R}^d$ is defined as $\gamma_{t;A} = \max_{\boldsymbol{X} := (\boldsymbol{x}_1 \ldots, \boldsymbol{x}_t) \in A^t} I(\boldsymbol{f_X}, \boldsymbol{y_X})$, where $I(\boldsymbol{f_X}, \boldsymbol{y_X}) := 0.5 \ln \det \left( \boldsymbol{I}_t + \sigma^{-2} \boldsymbol{K}(\boldsymbol{X}) \right)$ is the mutual information between the latent function values $\boldsymbol{f_X} \sim \mathcal{N}(\boldsymbol{0}, \boldsymbol{K}(\boldsymbol{X}))$ and corresponding output values $\boldsymbol{y_X} = \boldsymbol{f_X} + \boldsymbol{\epsilon}_t, \boldsymbol{\epsilon}_t \sim \mathcal{N}(\boldsymbol{0}, \sigma^2 \boldsymbol{I}_t)$ of GP. In GP-based optimization, $\gamma_{t;A}$ is often used as a quantity that characterizes the confidence bound and regret bound of $f$. Furthermore, for a compact convex set $A$, the upper bound for $\gamma_{t,A}$ has been derived for some commonly used kernels. For example, $\gamma_{t;A} = \mathcal{O}((\ln t)^{d+1})$ for Gaussian kernel and $\gamma_{t;A} = \mathcal{O}(t^{d/(\nu+d)}(\ln t)^{2\nu/(\nu+d)})$ for Matérn kernel with the smoothness parameter $\nu > 1/2$ [42, 49]. The following Lemma 2.1 adapts the well-known result that gives the confidence bound of $f$ for our problem setup, which is a direct consequence of Theorem 3.11 in [1].

**Lemma 2.1.** *Fix $f \in \mathcal{H}_k$ with $\|f\|_{\mathcal{H}_k} \leq B$. Suppose the observation $y_t = f(\boldsymbol{x}_t) + \epsilon_t$ has a noise $\epsilon_t$ that is conditionally $\sigma$-sub-Gaussian. Define $\{\beta_t\}_{t \in \mathbb{N}}$ as $\beta_t^{1/2} = B + \sigma \sqrt{2(\gamma_{t-1;\mathcal{X}} + 1 + \ln(1/\delta))}$ for $\delta \in (0, 1)$. Then the following holds with probability at least $1 - \delta$:*

$$\forall t \geq 1, \ \forall \boldsymbol{x} \in \mathcal{X}, \ \mathrm{lcb}_t(\boldsymbol{x}) \leq f(\boldsymbol{x}) \leq \mathrm{ucb}_t(\boldsymbol{x}). \tag{5}$$

*Here, $\mathrm{lcb}_t(\boldsymbol{x})$ and $\mathrm{ucb}_t(\boldsymbol{x})$ are defined as $\mathrm{lcb}_t(\boldsymbol{x}) = \mu_{t-1}(\boldsymbol{x}) - \beta_t^{1/2} \sigma_{t-1}(\boldsymbol{x})$ and $\mathrm{ucb}_t(\boldsymbol{x}) = \mu_{t-1}(\boldsymbol{x}) + \beta_t^{1/2} \sigma_{t-1}(\boldsymbol{x})$, respectively.*

---

[1] Some applications prefer to assess the performance via cumulative regret. In Appendix C, the analysis of the cumulative regret of our algorithm is provided.

**Regularity assumption for failure function.** A regret upper bound cannot be obtained without any assumption on $c$. As an extreme example, we consider the case where $\boldsymbol{x}^*$ is an isolated point surrounded by $F_c$. Under this scenario, the worst case exists in which an arbitrary algorithm can never observe $\boldsymbol{x}^*$ in a finite number of trials. Therefore, in order to give a convergence guarantee, at least $\boldsymbol{x}^*$ must be contained in a subset of the feasible region $S_c$ having a non-zero volume. In this paper, we focus on the case that $\boldsymbol{x}^*$ is the interior point of $S_c$ as in the following assumption.

**Assumption 2.2.** *There exists $\eta > 0$ such that $N_{\boldsymbol{x}^*;\eta} \subset S_c$, where $N_{\boldsymbol{x}^*;\eta} := \{\boldsymbol{x} \in \mathcal{X} \mid \|\boldsymbol{x} - \boldsymbol{x}^*\|_\infty < \eta\}$ is an open infinity ball with a radius $\eta$ centered at $\boldsymbol{x}^*$* [2].

The parameter $\eta$ above depends on the size of the subset of the feasible region that $\boldsymbol{x}^*$ belongs to and is an important quantity for theoretical analysis. Note that Assumption 2.2 is quite mild. For example, when $c$ is defined as $c(\boldsymbol{x}) = \mathbb{1}\{g(\boldsymbol{x}) \geq 0\}$ with a continuous latent function $g$, there exists $\eta > 0$ such that $N_{\boldsymbol{x}^*;\eta} \subset S_c$ from the continuity of $g$. Since the analysis of [2] assumes the existence of the latent function $g$, which is a *continuous* sample path generated from GP, our assumption also subsumes their assumption. Finally, note that $\eta$ is not needed for running the algorithm and is only used as a quantity that characterizes the regret bound.

## 3 Proposed algorithm

Our proposed algorithm F-GP-UCB is shown as pseudo-code in Algorithm 1. Roughly speaking, our F-GP-UCB searches the input domain excluding the adaptively adjusted neighborhood of past failure points based on the existing GP-UCB [42] strategy. Below, we start by describing the background of the algorithm construction.

**Philosophy of algorithm construction.** Since $F_c$ is unknown, the algorithm needs to appropriately avoid the failure observation in $F_c$ while balancing the trade-off between exploration and exploitation in a way that guarantees convergence. The difficulty of this problem lies in the fact that there exist cases where the small feasible region containing $\boldsymbol{x}^*$ is surrounded by the past failure observations and is isolated as shown in the left plot of Fig. 1. Since the algorithm is unable to exclude the possibility that an arbitrary small feasible region exists between failure points only from Assumption 2.2, the algorithm should be constructed so that it simultaneously satisfies the following two points: the failure observation should be avoided; and the inputs evaluated by the algorithm should eventually cover the arbitrarily close area of the past failure point.

F-GP-UCB satisfies the above requirements by adaptively controlling the GP-UCB search region by eliminating the neighborhood of the past failure points, and shrinking the eliminated region as step increase. Next, we describe below the details of and ideas behind each step.

**Selection of $\boldsymbol{x}_t$.** At each step $t$, the F-GP-UCB algorithm firstly computes the seach region $\mathcal{X}_t$ as

$$\mathcal{X}_t = \{\boldsymbol{x} \in \mathcal{X} \mid \forall i \in \mathcal{I}_{t-1}^{(F)}, \|\boldsymbol{x}_i - \boldsymbol{x}\|_\infty \geq \theta_t b(t)\}. \tag{6}$$

Here, $\theta_t$ and $b : \mathbb{R}_+ \to \mathbb{R}_+$ are the scale parameter and monotonically decreasing function, respectively. The parameter $\theta_t$ is controlled by the algorithm at every step so that it decreases monotonically with respect to $t$ from its initial value $\theta_0$ given by the user. It plays the role of guaranteeing that the search space $\mathcal{X}_t$ satisfies $\mathcal{X}_t \neq \emptyset$. The function $b$ is defined by the user before running the algorithm.

By using $\mathcal{X}_t$, the F-GP-UCB algorithm chooses $\boldsymbol{x}_t$ as

$$\boldsymbol{x}_t = \arg\max_{\boldsymbol{x} \in \mathcal{X}_t} \mathrm{ucb}_t(\boldsymbol{x}). \tag{7}$$

The middle and right plots in Fig. 1 show an example behavior of the F-GP-UCB algorithm. By using the monotonically decreasing $\theta_t b(t)$ to control the search space, the algorithm's behavior can be qualitatively described as follows. First, during the early phase where $t$ is small, $\theta_t b(t)$ is large, corresponding to choosing an unexplored point for observation while avoiding a large neighborhood of the past failure points where there is a high possibility of failure. Then as $t$ increases and the

---

[2]Our algorithm and theoretical analysis can be generalized for arbitrary norms over $\mathbb{R}^d$ such as $L_2$-norm and for a compact convex subset $\mathcal{X} \subset \mathbb{R}^d$, however the computational technique discussed in Sec.5 assumes the infinity norm.

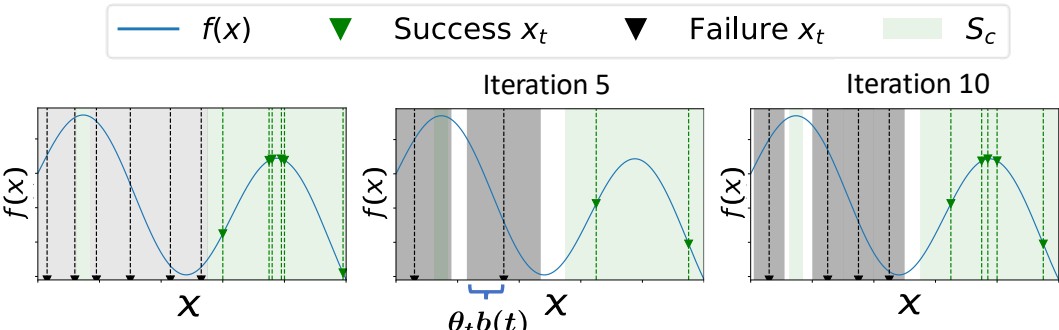

Figure 1: An example problem in one dimension. The left plot shows the situation where there exists an isolated feasible region, which includes the optimum. The green and grey shaded areas represent the feasible region $S_c$ and the failure region $F_c$, respectively. The green (black) points represent the observed successful (failure) points. In this situation, it is necessary to identify the feasible region which hides among the observed failure points. The middle and right plots are example behaviors of the F-GP-UCB algorithm in $t = 5$ and $t = 10$, respectively. The gray shaded regions represent the neighborhood of the observed failure points which is excluded from the search space at the given step. The F-GP-UCB algorithm performs searches in all the feasible regions while avoiding the observed failure points by iteratively narrowing down the excluded search space in the neighborhood of the failure points.

remaining unexplored space shrinks, it is expected that the algorithm's behavior will become more aggressive in considering the possibility that the region near the observed failure points contains a feasible region that may be difficult to identify.

The choice of $\theta_0$ and $b$ affects the performance of F-GP-UCB. In Sec. 4, we show the convergence of the regret under an appropriate choice of $b$. In Sec. 5, the practical choices of $\theta_0$ and $b$ are discussed.

**Shrinking $\theta_t$.** In the case where $\mathcal{X}_t = \emptyset$, $\boldsymbol{x}_t$ is not defined. We ensure such cases do not occur by controlling the scale parameter $\theta_t$. In our algorithm, the main role of $\theta_t$ is to guarantee that $\boldsymbol{x}_t$ is well-defined as follows. Specifically, before the step $t$ starts, $\theta_t$ is computed by halving the previous scale parameter $\theta_{t-1}$ until the union of all the neighborhoods does not fully cover $\mathcal{X}$. This procedure requires that we solve the set covering problem to find the cover of $\mathcal{X}$ with the neighborhoods of the past failure points. Unfortunately, this problem is known to be NP complete [20]. In Sec. 5, we provide a practical approach to this problem. On the other hand, note that our analysis in Sec. 4 assumes that this procedure can be computed exactly.

---

**Algorithm 1** The F-GP-UCB algorithm

**Input:** $\theta_0 \in (0, 1)$, $b : \mathbb{R}_+ \to \mathbb{R}_+$, $\{\beta_t\}_{t \in \mathbb{N}_+}$.

1: Initialize GP prior and set $\mathcal{I}_0^{(S)} = \mathcal{I}_0^{(F)} = \emptyset$.
2: **for** $t = 1$ to $T$ **do**
3:     $\tilde{\theta}_t \leftarrow \theta_{t-1}$.
4:     **while** $\mathcal{X} \subset \bigcup_{i \in \mathcal{I}_t^{(F)}} N_{\boldsymbol{x}_i; \tilde{\theta}_t b(t)}$ **do**
5:         $\tilde{\theta}_t \leftarrow \tilde{\theta}_t / 2$.
6:     **end while**
7:     $\theta_t \leftarrow \tilde{\theta}_t$ and define $\mathcal{X}_t$ as in (6).
8:     Choose $\boldsymbol{x}_t = \arg\max_{\boldsymbol{x} \in \mathcal{X}_t} \mathrm{ucb}_t(\boldsymbol{x})$.
9:     **if** $c(\boldsymbol{x}_t) = 0$ **then**
10:         Observe $y_t = f(\boldsymbol{x}_t) + \epsilon_t$ and update GP.
11:         $\mathcal{I}_t^{(S)} \leftarrow \mathcal{I}_{t-1}^{(S)} \cup \{t\}, \mathcal{I}_t^{(F)} \leftarrow \mathcal{I}_{t-1}^{(F)}$.
12:     **else**
13:         $\mathcal{I}_t^{(S)} \leftarrow \mathcal{I}_{t-1}^{(S)}, \mathcal{I}_t^{(F)} \leftarrow \mathcal{I}_{t-1}^{(F)} \cup \{t\}$.
14:     **end if**
15: **end for**
16: Calculate $\boldsymbol{x}_{\hat{T}}$ as in (8) if $\mathcal{I}_T^{(S)} \neq \emptyset$.

---

**Algorithm estimated solution.** The estimated solution of the F-GP-UCB algorithm $\hat{\boldsymbol{x}}_t$ at step $t$ is defined as follows:

$$\hat{\boldsymbol{x}}_t = \boldsymbol{x}_{\hat{t}} \quad \text{where} \quad \hat{t} = \arg\max_{i \in \mathcal{I}_t^{(S)}} \mathrm{lcb}_i(\boldsymbol{x}_i). \tag{8}$$

The definition of the estimated solution based on the observed points and $\mathrm{lcb}_i$ is often used in exisiting literature [6, 26, 21]. While the estimated solution of the proposed method resembles those in the

literature, it should be noted that the maximum of $\text{lcb}_i$ is computed over $\mathcal{I}_t^{(S)}$. Note $\hat{\boldsymbol{x}}_t$ is not defined if all the points up to $t$ are failures (i.e., $\mathcal{I}_t^{(S)} = \emptyset$). In Sec. 4, it is shown that there always exists $\hat{\boldsymbol{x}}_T$ for a sufficiently large step size $T$. Finally, although the estimated solution (8) is useful for discussing the theoretical performance, for practical purposes, $\hat{\boldsymbol{x}}_t = \boldsymbol{x}_{\hat{t}}$ where $\hat{t} = \text{argmax}_{i \in \mathcal{I}_t^{(S)}} \text{lcb}_t(\boldsymbol{x}_i)$ is often used instead [6, 21]. We use this modified definition of $\hat{\boldsymbol{x}}_t$ in the benchmark experiments in Sec. 6.

# 4 Theoretical analysis

The results of the theoretical analysis are discussed in this section. The complete proofs of the theorems and lemmas are described in Appendix A.

**Key of regret analysis.** To analyze the regret, we start by evaluating the growth rate of the number of successful observations $n_t$ in the algorithm. By noting that $\theta_t b(t)$ is monotonically decreasing, we can find that the failure points of F-GP-UCB are separated by at least a distance $\theta_t b(t)$ apart from each other. Therefore, $n_t$ can be evaluated via $\theta_t b(t)$-*packing number* of $F_c$. The following Lemma 4.1 is based on the standard argument for the packing number, which gives the lower bound for the number of successful observations $n_t$.

**Lemma 4.1.** *When running Algorithm 1, $n_t \geq t - \mathcal{P}(\mathcal{X}, \|\cdot\|_\infty, \theta_t b(t)/4)$ holds for any $t \in \mathbb{N}_+$, where $\mathcal{P}(\mathcal{X}, \|\cdot\|_\infty, \epsilon)$ denotes the $\epsilon$-packing number over the set $\mathcal{X}$ with respect to the norm $\|\cdot\|_\infty$.*

Using the above Lemma 4.1, designing the choice of $b$ so that $t - \mathcal{P}(\mathcal{X}, \|\cdot\|_\infty, \theta_t b(t)/4) = \Theta(t)$ holds leads to $n_t = \Omega(t)$, which means that the number of successful observations can be secured at the same rate as in the standard GP optimization setting. By using the fact $\mathcal{P}(\mathcal{X}, \|\cdot\|_\infty, \epsilon) = \lceil 1/\epsilon \rceil^d$ (Lemma A.4), the choice of $b(t) = o(t^{-1/d})$ is sufficient for $n_t = \Omega(t)^3$. While our result holds for any $b$ satisfying $b(t) = o(t^{-1/d})$, we give results for $b$ satisfying $b(t) = t^{-\alpha}$ with $\alpha \in (0, 1/d)$ in this section for simplicity. In Appendix A, the result for general $\mathcal{X}$ and $b$ is also provided.

**Regret bound.** The following Theorem 4.2 gives the regret bound of the F-GP-UCB algorithm.

**Theorem 4.2.** *Fix $\delta \in (0,1)$, $\alpha \in (0, d^{-1})$, $\mathcal{X} = [0,1]^d$, and $f \in \mathcal{H}_k$ such that $\|f\|_{\mathcal{H}_k} \leq B$. Let $\beta_t^{1/2} = B + \sigma\sqrt{2(\gamma_{t-1;\mathcal{X}} + 1 + \ln(1/\delta))}$ and $b(t) = t^{-\alpha}$. In addition, suppose that there exists $\eta > 0$ such that $N_{\boldsymbol{x}^*;\eta} \subset S_c$, where $N_{\boldsymbol{x}^*;\eta} = \{\boldsymbol{x} \in \mathcal{X} \mid \|\boldsymbol{x} - \boldsymbol{x}^*\|_\infty < \eta\}$. When applying Algorithm 1 under the above conditions, the estimated solution $\hat{\boldsymbol{x}}_T$ defined in (8) exists for all $T \geq \tilde{T}_s := \max\{(10/\overline{\theta}^d)^{1/(1-d\alpha)}, 2\} + 1$, where $\overline{\theta} = \min\{\theta_0, \eta/2\}$. Moreover, the following holds with probability at least $1 - \delta$:*

$$\forall T \geq \tilde{T}_s, \; r_T \leq \frac{2}{T}\left[2B(\tilde{T}_* - 1) + \sqrt{C_1 \beta_T T \gamma_{T;S_c}}\right], \tag{9}$$

*where $\tilde{T}_* = (\theta_0/\eta)^{1/\alpha} + 1$ and $C_1 = 8/\ln(1 + \sigma^{-2})$.*

In the analysis above, $\eta$ is interpreted as the complexity of the failure function $c$, and $\overline{\theta}$ corresponds to the lower bound on $\theta_t$ while running the algorithm. The constants $\tilde{T}_s$ and $\tilde{T}_*$ are important quantities that characterize the regret and can be respectively interpreted as follows.

- The constant $\tilde{T}_s$ is the upper bound on the number of steps until the first successful observation.

- The constant $\tilde{T}_*$ is the upper bound on the number of steps needed until the search space $\mathcal{X}_t$ of the algorithm always contains the optimal solution $\boldsymbol{x}^*$.

The regret bound in Theorem 4.2 can be intuitively seen as evaluating the algorithm execution in terms of two stages along the time axis. The first term of (9) represents the regret generated when the algorithm's search space $\mathcal{X}_t$ does not contain $\boldsymbol{x}^*$. Specifically, the worst-case regret incurred during this phase is bounded by $2B$ using the fact that $\sup_{\boldsymbol{x} \in \mathcal{X}} |f(\boldsymbol{x})| \leq B$ from the kernel normalization

---

[3]Strictly speaking, we use the fact that $\theta_t$ becomes constant for sufficiently large $t$. See Lemma A.8 in Appendix A.

condition. The second term is the regret incurred during the process of identifying $x^*$ after $\mathcal{X}_t$ includes $x^*$ and is similar to the regret bound of standard GP-UCB. The only difference is that, in our setup, the observations of GP are only on $S_c$, so the maximum information gain is defined over $S_c$.

By noting that $\gamma_{T;S_c} \subset \gamma_{T;\mathcal{X}}$ in the second term and using the known results concerning the upper bound on $\gamma_{T;\mathcal{X}}$, the more explicit form of regret upper bound can be obtained on a per-kernel basis. For example, in the case of the Gaussian kernel, we have that $\gamma_{T;\mathcal{X}} = \mathcal{O}((\ln T)^{d+1})$ and $r_T = \mathcal{O}\left(T^{-1}B\tilde{T}_* + T^{-1/2}(\ln T)^{(d+1)/2}\left(B + \sigma\sqrt{(\ln T)^{d+1} + \ln(1/\delta)}\right)\right)$. Thus, indeed we obtain $r_T \to 0$ (as $T \to \infty$). The F-GP-UCB algorithm is guaranteed to converge when the cumulative regret for ordinary GP-UCB (corresponding to the second term in (9)) becomes sub-linear under the chosen setup and the kernel[4].

**The case where $x^*$ exists on the boundary of $S_c$.** The case where $x^*$ exists on the boundary $S_c$ is not covered in Theorem 4.2, which is the case that often appears in a real-world application. If undesirable Lipschitz-style dependencies are allowed to emerge in the regret upper bound, our F-GP-UCB algorithm is also guaranteed to have convergence in such cases (Theorem B.2). The details of the result and discussion about its limitation are in Appendix B.

## 5 Practical considerations

In this section, we discuss several issues that arise in practical situations and provide their solutions. Due to the space limitation, we only give brief descriptions. The details are described in Appendix E.

**Computation of $\theta_t$.** The computation of $\theta_t$ requires us to know whether the union of all the neighborhoods of the failure points covers $\mathcal{X}$. A simple solution might be to partition $\mathcal{X}$ into a sufficiently fine grid by subdividing each axis and look for a feasible solution in each grid cell by brute force. However, as the step size $t$ and dimension $d$ become large, the number of grid cells grows rapidly. Furthermore, even a very fine grid will not be able to detect the case where a feasible solution exists on the cell boundary. Another heuristic approach is to use a constrained optimization solver which does not require a feasible solution as part of the initial points (e.g., penalty function method and augmented Lagrangian method [17]) and try to solve the problem (7) by using $\theta_{t-1}$. In case that the solver is unable to find a feasible solution, it is decided that $\tilde{\mathcal{X}}_t \neq \emptyset$, multiply $\theta_{t-1}$ by $1/2$, then we try to solve problem (7) again. This is our chosen approach. A drawback of this approach is that, depending on the scale of $\theta_t$, the constrained optimization problem must be solved many times. We show that, by performing an appropriate preprocessing, a feasible solution can be obtained within two rounds of constrained optimization in the worst case. We give the details in Appendix E.1.

**Selection of $b$.** As stated in Sec. 4, the F-GP-UCB algorithm is valid for an arbitrary monotonically decreasing function $b$ such that $t - \mathcal{P}(\mathcal{X}, \|\cdot\|_\infty, \theta_t b(t)/4) = \Theta(t)$. However, if the rate is slower than $1/t^\alpha$ ($\alpha \in (0, 1/d)$), for example, $b(t) = 1/\ln(t+1)$, then the dominant term of our regret upper bound can become $\mathcal{O}(1/\ln T)$ (See Theorem B.3 in Appendix B and Corollary F.1 in Appendix F). This intuitively means that an exponentially large sample is needed to obtain an arbitrarily small regret and is not the desirable behavior. In this paper, we use $b$ in the form of $b(t) = 1/t^\alpha$ with $\alpha = 1/(2d)$ as a practical choice. (Additional discussion about $\alpha$ is given in Appendix E.2.)

**Adaptive tuning of $\theta_0$.** Although the choice of $\theta_0$ does not affect the no-regret guarantee, it has a large impact on the practical behavior when the number of steps is small. For example, if $\theta_0$ is set to a large value when the failure region $F_c$ forms a relatively small region, then there is a risk of excessively avoiding the feasible region that should be searched. It is practically difficult to find a desirable choice of $\theta_0$ as it depends on some conditions that are unknown to the user, such as the failure region $F_c$. In order to alleviate this issue while retaining the theoretical convergence, we use the strategy to adaptively choose $\theta_t$ based on the posterior standard deviation $\sigma_{t-1}(x_t)$ of the observed points. We note that such a strategy using $\sigma_{t-1}(x_t)$ to adaptively select an unknown parameter is also developed in [54] to set the unknown kernel hyperparameters for ordinary GP optimization.

---

[4]As with the standard GP-UCB, $\beta_t = \mathcal{O}(\sqrt{\gamma_{T;\mathcal{X}}})$ in F-GP-UCB, which leads to the regret of $\tilde{\mathcal{O}}(\gamma_{T;\mathcal{X}}/\sqrt{T})$. In Appendix D, we discuss the possibility of addressing this issue based on several existing works [48, 30].

Algorithm 4 in Appendix E.3 shows the pseudo-code of our modified strategy. In Algorithm 4, the user specifies the possible range for $\theta_t \in [\theta_{\min}, \theta_{\max}]$, the threshold for the posterior standard deviation $h_\sigma > 0$, the threshold for the step number $q \in \mathbb{N}_+$, and a scaling factor $w \in (0, 1)$. As the initial value, we set $\theta_0 = \theta_{\max}$. After that, if the observed point is lower than the posterior standard deviation for $q$ consecutive number of times, $\theta_t$ is multiplied by $w$ to lower its value, but no less than the minimum value $\theta_{\min}$. Intuitively, suppose the posterior standard deviation is repeatedly excessively small. In that case, there exists a possibility that $\theta_0$ was set to an excessively large value, which may lead to the unwanted exclusion of the feasible region from the search space. In this procedure, by setting $\theta_0$ to a large value of $\theta_{\max}$ at the start, we expect it to get adjusted towards an appropriate neighborhood scale. In our simulation experiments, we set $w = 0.75$, $h_\sigma = 0.02$, $q = 3$, $\theta_{\min} = 0.0001$, and $\theta_{\max} = 0.5$. We emphasize that this procedure does not affect our convergence guarantee (Theorems F.2 in Appendix E).

Lastly, the pseudo-code, including all the considerations provided in this section, is described in Algorithm 5 of Appendix E.

## 6   Numerical experiments

In this section, we show the performance of F-GP-UCB through numerical experiments. The detailed settings of the experiments and additional results can be found in Appendices G and H, respectively. First, for comparison, we use EI [37] and GP-UCB [42] as the baseline algorithms which do not consider failures. In each of these algorithms, the GP model is constructed using only the successful observations; the failure observations are not used. We also compare F-GP-UCB with the EFI-GPC-EP and EFI-GPC-Sign algorithms that leverage GPC models [2]. EFI-GPC-EP is the algorithm that uses the classic GPC model with probit likelihood, and posterior approximation is based on Expectation Propagation (EP). EFI-GPC-Sign uses a variant of GPC proposed in [2]. Both EFI-GPC-EP and EFI-GPC-Sign choose the next input based on the posterior success probability of GPC and the EI value of $f$ as in [2] (Details are in Appendix H).

In EFI-GPC-EP and EFI-GPC-Sign, as the prior for the latent function of GPC, we use zero-mean GP with Gaussian kernel $k_c(\boldsymbol{x}, \boldsymbol{y}) \coloneqq \sigma_c^2 \exp(-\|\boldsymbol{x} - \boldsymbol{y}\|^2/(2l_c^2))$. Before the experiment, we fix the kernel parameters of GPC $\sigma_c$ and $l_c$ by marginal likelihood maximization using observations at a randomly generated Sobol sequence. All the kernel hyperparameters of the GPC fixed by this procedure are given in Appendix H. Furthermore, it is well-known that the theoretical choice of $\beta_t$ in GP-UCB is excessively conservative. We set $\beta_t = 2\ln(2t)$ as in [24] for GP-UCB and F-GP-UCB. The other parameters in F-GP-UCB are fixed as described in Sec. 5. For the evaluation of the algorithms, we confirm the behavior of the regret at each step $r_t = f(\boldsymbol{x}^*) - f(\hat{\boldsymbol{x}}_t)$. We note that $\hat{\boldsymbol{x}}_t$ is the estimated solution $\hat{\boldsymbol{x}}_t = \boldsymbol{x}_{\hat{t}}$ where $\hat{t} = \mathrm{argmax}_{i \in \mathcal{I}_t^{(S)}} \mathrm{lcb}_t(\boldsymbol{x}_i)$, which is slightly modified from the theoretical value as discussed in Sec. 3 If we get that $\mathcal{I}_t^{(S)} = \emptyset$, then we set $r_t = f(\boldsymbol{x}^*) - \min_{\boldsymbol{x} \in \mathcal{X}} f(\boldsymbol{x})$. This definition is the same as the utility gap metric for constrained GP optimization [19]. In each experiment, we report the average performance of 20 numbers of optimization trials with different seeds. In each trial, we generate one point uniformly at random over $\mathcal{X}$ and use it as the initial point. Lastly, we add an artificial noise $\epsilon_t \sim \mathcal{N}(0, \sigma^2)$ with $\sigma^2 = 0.0001$ to the observation of $f$. We also fix the noise variance hyperparameter of the GP model for $f$ to $0.0001$.

**Synthetic experiments with the Branin function.**   We perform synthetic benchmark experiments with the Branin function whose input space is scaled to $[0, 1]^2$. The failure function used in these experiments is shown in Fig. 2. This failure function has a feasible region on the upper right side which is easily identifiable. In addition, the failure function also contains multiple small isolated feasible regions, one of which contains the optimal solution. We use the Gaussian kernel $k(\boldsymbol{x}, \boldsymbol{y}) \coloneqq \sigma_f^2 \exp(-\|\boldsymbol{x} - \boldsymbol{y}\|_2^2/(2l_f^2))$ to construct a GP model of $f$. The parameters $\sigma_f$ and $l_f$ are fixed beforehand by marginal likelihood maximization over a Sobol sequence of $1024$ points. The subsequent experiments also use the Gaussian kernel; and the parameters are fixed as described above.

Figure 2 shows the result. Under this benchmark setting, F-GP-UCB has a better performance. We also give the detailed behavior of F-GP-UCB in Appendix G. We note that, in EI and GP-UCB, the regret stops decreasing in the early stage. This is because these algorithms get stuck on the same failure point.

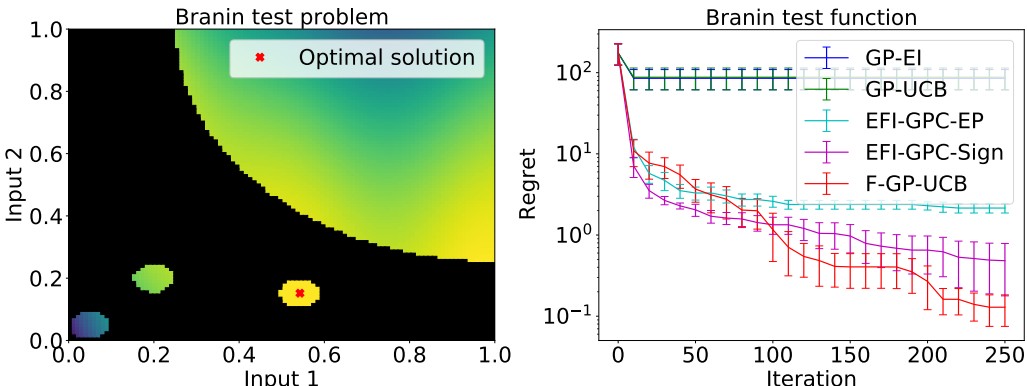

Figure 2: The left plot shows the Branin function whose input space is scaled to $[0, 1]^2$. The shaded regions represent the failure regions. The right plot shows the regret in the synthetic problem using the Branin function, where the average of 20 experiments with different random seeds is shown. The error bars correspond to two standard errors.

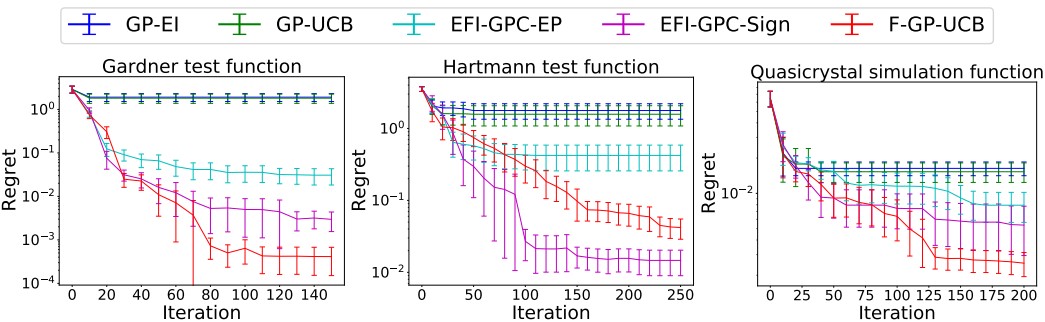

Figure 3: Experiment results of benchmark function of the constrained optimization problems (left and middle) and quasicrystal simulation function (right). The left and middle plots show the result of Gardner and Hartmann test functions, respectively. The error bars correspond to two standard errors.

**Test functions for constrained optimization.** We adopt the two benchmark settings which are used in existing GP-based constrained optimization literature. Gardner [10] is a 2D test problem whose objective and constraint functions are defined as a combination of sine and cosine functions. Hartmann [29] is a 3D test problem[5] whose feasible region becomes a unit hypersphere. To adapt the constrained problem to our settings, the feasible region in each benchmark is defined as the success region, while the rest is defined as the failure region. Fig. 3 shows the result. Within the compared algorithms, F-GP-UCB has the best performance for the Gardner function, while EFI-GPC-Sign has the best performance for the Hartmann function. This can be understood from the fact that the feasible region of the Hartmann function, which is a unit hypersphere, and can be modeled easily by GPC. However, F-GP-UCB performs better compared to EFI-GPC-EP, which uses a classic GPC model. Moreover, the regret of F-GP-UCB continues to decrease as the number of iterations increases.

**Numerical experiments with quasicrystals.** We perform a numerical experiment involving quasicrystals in the Al-Cu-Mn system. We consider the optimization of the phonon thermal conductivity for quasicrystals. In this setting, the input space is the relative composition of the three elements, restricted to be around the feasible region of quasicrystal formation. The feasible region is the composition values that form quasicrystals based on data [16]. Outside of this region, quasicrystals do not form and so we consider it as a failure region, as the material property could not be probed with methods designed for quasicrystals. For the objective function, we use an empirical formula for the phonon thermal conductivity which we extract from [46]. The details of the setting are described in

---

[5]The original paper [29] considers a 6D problem and this setting leads to an excessively large failure region. We modify the setting by using a 3D version of the Hartmann function with unit hypersphere constraint.

Appendix H.2. In Fig. 3, the right plot shows the result of the numerical experiments for quasicrystals. It shows that F-GP-UCB performs the best among the compared algorithms.

## 7 Conclusions

In this paper, we propose a novel GP-based optimization algorithm in the presence of unknown failure regions in the search space. Our algorithm only requires a very mild assumption of the failure function that the optimal solution lies on an interior of a feasible region. We show that our algorithm achieves a convergence with high probability and provides the first regret upper bound by appropriately adjusting the search space. Its effectiveness is verified empirically through numerical experiments, including the heuristic simulation experiment motivated by the material research of quasicrystals.

## Acknowledgements

This work was partially supported by RIKEN Center for Advanced Intelligence Project.

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
