# A Proofs of Section 4

In this section, we prove the regret bound of the proposed algorithm for the case of an arbitrary normed space $\mathcal{X}$. This is followed by proofs of Theorems 4.2 and B.2.

## A.1 Preliminaries for proofs

We first define the packing number and the covering number and provide some known inequalities for those.

**Definition A.1** (Packing number). Let $(\mathcal{A}, \| \cdot \|)$ be a normed space and $\epsilon > 0$. If a subset $\mathcal{N} \subset \mathcal{A}$ satisfies the following equation, we say that $\mathcal{N}$ is an $\epsilon$-separated set:

$$\|\boldsymbol{x} - \boldsymbol{y}\| > \epsilon \text{ for all distinct } \boldsymbol{x}, \boldsymbol{y} \in \mathcal{N}. \tag{10}$$

Furthermore, we define $\mathcal{P}(A, \| \cdot \|, \epsilon)$ with respect to $A \subset \mathcal{A}$ as the maximum possible number of elements of the $\epsilon$-separated set $\mathcal{N} \subset A$ of $A$.

**Definition A.2** (Covering number). Let $(\mathcal{A}, \| \cdot \|)$ be a normed space, $A \subset \mathcal{A}$ be its subset, and $\epsilon > 0$. If a subset $\mathcal{N} \subset \mathcal{A}$ satisfies the following equation, we say that $\mathcal{N}$ is an $\epsilon$-net of $A$:

$$\forall \boldsymbol{x} \in \mathcal{X}, \exists \boldsymbol{y} \in \mathcal{N}, \|\boldsymbol{x} - \boldsymbol{y}\| \leq \epsilon. \tag{11}$$

Furthermore, we define $\mathcal{C}(A, \| \cdot \|, \epsilon)$ to be the minimum possible number of elements of the $\epsilon$-net $\mathcal{N} \subset \mathcal{A}$ of $A$.

**Lemma A.3.** *Let $(\mathcal{A}, \| \cdot \|)$ be an arbitrary normed space, $A \subset \mathcal{A}$ be its subset, and $\epsilon > 0$. Then the following holds:*

- $\mathcal{P}(A, \| \cdot \|, 2\epsilon) \leq \mathcal{C}(A, \| \cdot \|, \epsilon) \leq \mathcal{P}(A, \| \cdot \|, \epsilon)$.

- $\tilde{A} \subset A \Rightarrow \mathcal{C}(\tilde{A}, \| \cdot \|, \epsilon) \leq \mathcal{C}(A, \| \cdot \|, \epsilon/2)$.

The proof of Lemma A.3 can be found, e.g., around Lemma 4.2.8 in Ch. 4 of Vershynin [52].

Then, we show the packing number of the unit hypercube:

**Lemma A.4.** *Let $\epsilon > 0$ and $d \in \mathbb{N}_+$. Then the following holds:*

$$\mathcal{P}\left([0, 1]^d, \| \cdot \|_\infty, \epsilon\right) = \left\lceil \frac{1}{\epsilon} \right\rceil^d. \tag{12}$$

*Proof.* Let $W = \{v/2 + j\epsilon \mid j \in \{0, 1, \ldots, u\}\}$ for $u \in \mathbb{N}_+$ and $v \in (0, \epsilon]$ such that $u\epsilon + v = 1$, and $\mathcal{N} \subset [0, 1]^d$ be an $\epsilon$-separated set such that $|\mathcal{N}| = \mathcal{P}\left([0, 1]^d, \| \cdot \|_\infty, \epsilon\right)$. We further denote $\boldsymbol{y_x} \in W^d$ as the nearest point in $W^d$ for an arbitrary $\boldsymbol{x} \in [0, 1]^d$, which satisfies $\|\boldsymbol{x} - \boldsymbol{y_x}\|_\infty \leq \epsilon/2$ from the definition of $W$. If the equality $\boldsymbol{y_{x_1}} = \boldsymbol{y_{x_2}}$ holds for some distinct elements $\boldsymbol{x}_1$ and $\boldsymbol{x}_2$ of $\mathcal{N}$, then it follows that

$$\|\boldsymbol{x}_1 - \boldsymbol{x}_2\|_\infty \leq \|\boldsymbol{x}_1 - \boldsymbol{y_{x_1}}\|_\infty + \|\boldsymbol{x}_2 - \boldsymbol{y_{x_2}}\|_\infty \tag{13}$$
$$\leq \epsilon, \tag{14}$$

which contradicts that $\mathcal{N}$ is the $\epsilon$-separated set. Therefore, $\boldsymbol{y_{x_1}} \neq \boldsymbol{y_{x_2}}$. Thus, there exists an injection from $\mathcal{N}$ to $W^d$, i.e., $|\mathcal{N}| \leq |W^d|$. Furthermore, if we let $\tilde{W} = \{j(\epsilon + v/(u + 1)) \mid j \in \{0, 1, \ldots, u\}\}$, then $\tilde{W}^d$ is an $\epsilon$-separated set and $|W^d| = |\tilde{W}^d| \leq |\mathcal{N}|$. It follows that $|W^d| = |\mathcal{N}| = \mathcal{P}\left([0, 1]^d, \| \cdot \|_\infty, \epsilon\right)$. From the definition of $W$, we obtain $|W^d| = \lceil 1/\epsilon \rceil^d$. This completes the proof. $\square$

## A.2 Proof of Theorem 4.2.

Theorem 4.2 assumes $\mathcal{X} = [0, 1]^d$, $b(t) = t^{-\alpha}$ with $\alpha \in (0, d^{-1})$, and the search region $\mathcal{X}_t$ which is defined based on the infinity norm $\| \cdot \|_\infty$. Before the proof of Theorem 4.2, we prove the theorem for Algorithm 1 that uses the general $\mathcal{X}$, $b(\cdot)$, and the norm $\| \cdot \|$, which satisfy the following Assumption A.5.

**Assumption A.5.** *Suppose that the input domain $\mathcal{X}$, the function $b(\cdot)$, and the search region $\mathcal{X}_t$ in the Algorithm 1 satisfy the following:*

- *$b : [1, \infty) \to (0, \infty)$ is a strictly monotonically decreasing continuous function such that $\lim_{t \to \infty} b(t) = 0$.*

- *$\mathcal{X}$ is a compact subset of an arbitrary normed space equipped with norm $\|\cdot\|$.*

- *The search region $\mathcal{X}_t$ at each step $t$ is given using the norm $\|\cdot\|$ of $\mathcal{X}$ as follows:*

$$\mathcal{X}_t = \{\boldsymbol{x} \in \mathcal{X} \mid \forall i \in \mathcal{I}_t^{(F)}, \|\boldsymbol{x}_i - \boldsymbol{x}\| \geq \theta_t b(t)\}. \tag{15}$$

In addition, we define $\tilde{\mathcal{X}}_t$ as the search space for $\theta_{t-1}$ instead of $\theta_t$. That is,

$$\tilde{\mathcal{X}}_t = \{\boldsymbol{x} \in \mathcal{X} \mid \forall i \in \mathcal{I}_t^{(F)}, \|\boldsymbol{x}_i - \boldsymbol{x}\| \geq \theta_{t-1} b(t)\}. \tag{16}$$

We also define the $\boldsymbol{x}_t^*$ as one of the optimal solutions over $\mathcal{X}_t$, i.e., $\boldsymbol{x}_t^* \in \operatorname{argmax}_{\boldsymbol{x} \in \mathcal{X}_t} f(\boldsymbol{x})$. We first prove Theorem A.6, which is the more general form of Theorem 4.2.

**Theorem A.6.** *Fix $\delta \in (0, 1)$ and $f \in \mathcal{H}_k$ such that $\|f\|_{\mathcal{H}_k} \leq B$. Let $\beta_t^{1/2} = B + \sigma \sqrt{2(\gamma_{t-1;\mathcal{X}} + 1 + \ln(1/\delta))}$. We further assume the following two conditions.*

1. *There exists $\eta > 0$ such that $N_{\boldsymbol{x}^*; \eta, \|\cdot\|} \subset S_c$ holds where $N_{\boldsymbol{x}^*; \eta, \|\cdot\|} := \{\boldsymbol{x} \in \mathcal{X} \mid \|\boldsymbol{x} - \boldsymbol{x}^*\| < \eta\}$.*

2. *$\lim_{T \to \infty} \left(T - \mathcal{P}\left(F_c, \|\cdot\|, \overline{\theta} b(T)\right)\right) = \infty$ holds for $\overline{\theta} = \min\{\theta_0, \eta/(2b(1))\}$.*

*Then, letting $s(T) = T - \mathcal{P}(F_c, \|\cdot\|, \overline{\theta} b(T))$, the following holds under Assumption A.5 in Algorithm 1:*

- *$\hat{\boldsymbol{x}}_T$ exists for all $T \in \mathbb{N}_+$ such that $T \geq T_s$.*

- *The following holds with probability at least $1 - \delta$:*

$$\forall T \geq T_s, r_T \leq \frac{1}{s(T)} \left[2B(T_* - 1) + \sqrt{C_1 \beta_T T \gamma_{T;S_c}}\right]. \tag{17}$$

*Here, $T_*$ is a natural number defined as follows:*

$$T_* = \begin{cases} \lfloor b^{-1}(\frac{\eta}{\theta_0}) + 1 \rfloor & \text{if } \frac{\eta}{\theta_0} \leq b(1), \\ 1 & \text{otherwise} \end{cases}. \tag{18}$$

*Moreover, we let $T_s$ be the smallest natural number $T$ such that $s(T) \geq 1$ and $C_1 = 8/\ln(1 + \sigma^{-2})$.*

The variables $\overline{\theta}$, $T_s$, $T_*$, and $s(T)$ are important quantities for regret analysis and can be interpreted as follows:

- $\overline{\theta}$ is the lower bound on $\theta_t$ for $t \in \mathbb{N}_+$.
- $T_s$ is the upper bound on the number of steps until the first successful observation.
- $T_*$ is the upper bound on the number of steps until $\boldsymbol{x}^* \in \mathcal{X}_t$ holds.
- $s(T)$ is the lower bound on the number of successful observations at step $T$.

Note that $\overline{\theta}$, $T_s$, and $T_*$ depend only on $c$, $\eta$, $\theta_0$, and $b$ and does not depend on $T$. We provide some lemmas before proving Theorem A.6.

**Lemma A.7.** *The following holds for any $t \in \mathbb{N}_+$:*

$$n_t \geq t - \mathcal{P}\left(F_c, \|\cdot\|, \theta_t b(t)\right). \tag{19}$$

*Proof.* The set of failure points $\{\boldsymbol{x}_i\}_{i \in \mathcal{I}_t^{(F)}}$ at step $t$ is a $\theta_t b(t)$-separated set over $F_c$ due to the monotonicity of $\theta_t b(t)$. It follows that

$$n_t = t - |\mathcal{I}_t^{(F)}| \tag{20}$$
$$\geq t - \mathcal{P}\left(F_c, \|\cdot\|, \theta_t b(t)\right). \tag{21}$$

$\square$

**Lemma A.8.** *The following holds for any natural number $t$ such that $t \geq T_*$:*

1. $\mathcal{X}_t$ *contains the optimal solution, i.e.,* $\boldsymbol{x}^* \in \mathcal{X}_t$.

2. $\theta_t = \theta_{T_*}$ *and* $\theta_{T_*} \geq \min\{\theta_0, \eta/(2b(1))\}$.

*Proof.* We first prove the first statement. Given an arbitrary $t \geq T_*$, the following holds from the definition of $T_*$:

- If $\eta/\theta_0 \leq b(1)$, then $t \geq \lfloor b^{-1}(\eta/\theta_0) + 1 \rfloor > b^{-1}(\eta/\theta_0) \Rightarrow \theta_0 b(t) < \eta$.

- If $\eta/\theta_0 > b(1)$, then $\theta_0 b(1) < \eta \Rightarrow \theta_0 b(t) < \eta$.

Thus, in either case, $\theta_0 b(t) < \eta$ holds. Combining it with the monotonicity of $\theta_t$, we have $\theta_t b(t) < \eta$. This implies $\boldsymbol{x}^* \in \mathcal{X}_t$.

Concerning the second statement, $\theta_t b(t+1) < \theta_t b(t) < \eta$ holds for $t \geq T_*$. Thus, we have $\boldsymbol{x}_* \in \tilde{\mathcal{X}}_{t+1}$. That is, $\tilde{\mathcal{X}}_{t+1} \neq \emptyset$ and $\theta_t = \theta_{t+1}$. It follows that $\theta_t$ is equal for all $t \geq T_*$ and $\theta_t = \theta_{T_*}$ holds.

Let $\theta_{i-1} > \theta_i$ and $\theta_i = \theta_{T_*}$ for some index $i < T_*$. Then, if $\theta_{i-1} b(i) < \eta$, we have that $\tilde{\mathcal{X}}_i \neq \emptyset$, which contradicts $\theta_{i-1} > \theta_i$. Thus, $\theta_{i-1} b(i) \geq \eta$. Note that reducing $\theta_{i-1}$ is done by multiplying $1/2$ and the reduction at least stops when $\tilde{\theta}_i b(i) < \eta$ holds for the newly reduced $\tilde{\theta}_i$, we have that $\theta_i b(i) \geq \eta/2$. It follows that

$$\theta_{T_*} = \theta_i \geq \frac{\eta}{2b(i)} > \frac{\eta}{2b(1)}. \tag{22}$$

Combining the fact that we have $\theta_{T_*} = \theta_0$ if the index $i$ does not exist, we can obtain $\theta_{T_*} \geq \min\{\theta_0, \eta/(2b(1))\}$, which completes the proof. $\square$

**Lemma A.9.** *Let $T$ be a natural number such that $n_t \geq 1$. Then the following holds:*

$$\sum_{t \in \mathcal{I}_T^{(S)}} \sigma_{t-1}(\boldsymbol{x}_t) \leq \sqrt{\frac{2n_t}{\ln(1 + \sigma^{-2})} \gamma_{n_t; S_c}} \tag{23}$$

Lemma A.9 immediately holds from the well-known inequality about the maximum information gain [42].

**Lemma A.10.** *The following holds for any $t \in \mathbb{N}_+$:*

$$n_t \geq t - \mathcal{P}\left(F_c, \|\cdot\|, \bar{\theta}b(t)\right). \tag{24}$$

*Proof.* From Lemma A.8, $\forall t \in \mathbb{N}, \theta_t \geq \bar{\theta}$. It follows that, together with the monotonicity of the packing number over the normed space, $\forall t \in \mathbb{N}_+, \mathcal{P}\left(F_c, \|\cdot\|, \theta_t b(t)\right) \leq \mathcal{P}\left(F_c, \|\cdot\|, \bar{\theta}b(t)\right)$. Using Lemma A.7 completes the proof. $\square$

We give the proof for Theorem A.6.

*Proof of theorem A.6.* For any natural number $T$ such that $T \geq T_s$, we have that $n_t \geq 1$ from Lemma A.10, and the definition of $T_s$. It follows that $\boldsymbol{x}_T$ exists. We show the regret bound next. We first assume that Eq. (5) holds. Here, we fix a natural number $T \geq T_s$. Then,

$$r_T = f(\boldsymbol{x}^*) - f(\hat{\boldsymbol{x}}_T) \tag{25}$$

$$= n_t^{-1} \sum_{t \in \mathcal{I}_T^{(S)}} \left(f(\boldsymbol{x}^*) - f(\boldsymbol{x}_t^*) + f(\boldsymbol{x}_t^*) - f(\hat{\boldsymbol{x}}_T)\right) \tag{26}$$

$$\leq s(T)^{-1} \left[ \sum_{t \in \mathcal{I}_T^{(S)}} \left(f(\boldsymbol{x}^*) - f(\boldsymbol{x}_t^*)\right) + \sum_{t \in \mathcal{I}_T^{(S)}} \left(f(\boldsymbol{x}_t^*) - f(\hat{\boldsymbol{x}}_T)\right) \right]. \tag{27}$$

From Lemma A.8, the first term in the equation above becomes $\boldsymbol{x}^* = \boldsymbol{x}_t^*$ for any natural number $t$ such that $t \geq T_*$. It follows that

$$\sum_{t \in \mathcal{I}_T^{(S)}} (f(\boldsymbol{x}^*) - f(\boldsymbol{x}_t^*)) \leq \sum_{t=1}^{T} (f(\boldsymbol{x}^*) - f(\boldsymbol{x}_t^*)) \tag{28}$$

$$\leq \sum_{t=1}^{T_*-1} (f(\boldsymbol{x}^*) - f(\boldsymbol{x}_t^*)) \tag{29}$$

$$\leq 2B(T_* - 1). \tag{30}$$

In the last line, we used the fact that $\sup_{\boldsymbol{x} \in \mathcal{X}} |f(\boldsymbol{x})| \leq \|f\|_{\mathcal{H}_k} \leq B$ due to $\forall \boldsymbol{x} \in \mathcal{X}, k(\boldsymbol{x}, \boldsymbol{x}) \leq 1$.

Concerning the second term, we have, for any $t \in \mathcal{I}_T^{(S)}$,

$$f(\boldsymbol{x}_t^*) - f(\hat{\boldsymbol{x}}_T) \leq \text{ucb}_t(\boldsymbol{x}_t) - \max_{i \in \mathcal{I}_T^{(S)}} \text{lcb}_i(\boldsymbol{x}_i) \tag{31}$$

$$\leq \text{ucb}_t(\boldsymbol{x}_t) - \text{lcb}_t(\boldsymbol{x}_t) \tag{32}$$

$$\leq 2\beta_t^{1/2} \sigma_{t-1}(\boldsymbol{x}_t). \tag{33}$$

Using Lemma A.9,

$$\sum_{t \in \mathcal{I}_T^{(S)}} (f(\boldsymbol{x}^*) - f(\hat{\boldsymbol{x}}_T)) \leq 2\beta_T^{1/2} \sum_{t \in \mathcal{I}_T^{(S)}} \sigma_{t-1}(\boldsymbol{x}_t) \tag{34}$$

$$\leq 2\beta_T^{1/2} \sqrt{\frac{2n_t}{\ln(1 + \sigma^{-2})} \gamma_{n_t; S_c}} \tag{35}$$

$$\leq \sqrt{\frac{8\beta_T T}{\ln(1 + \sigma^{-2})} \gamma_{T; S_c}} \tag{36}$$

$$\leq \sqrt{C_1 \beta_T T \gamma_{T; S_c}}. \tag{37}$$

In summary, assuming Eq. (5) holds,

$$\forall T \geq T_s, r_T \leq \frac{1}{s(T)} \left[ 2B(T_* - 1) + \sqrt{C_1 \beta_T T \gamma_{T; S_c}} \right]. \tag{38}$$

Finally, from Lemma 2.1, Eq. (5) holds with probability at least $1 - \delta$. This completes the proof. $\quad\square$

We consider $S_c = \mathcal{X}$ as a special case of Theorem A.6. Then, some $\eta$ exists such that $\theta_0 b(1) < \eta$ and we have $T_* = 1$. It follows that $s(T) = T$ and $T_s = 1$. This matches the standard regret bound for GP-UCB.

Furthermore, if we fix $b$, $\eta$, and $B$ and consider only the dependence on $T$, we obtain $r_T = \mathcal{O}(\sqrt{\beta_T T \gamma_{T; \mathcal{X}}}/s(T))$. (Here, we used $\gamma_{T; S_c} \leq \gamma_{T; \mathcal{X}}$ which follows from $S_c \subset \mathcal{X}$.) We note that the cumulative regret in standard GP-UCB is $\mathcal{O}(\sqrt{\beta_T T \gamma_{T; \mathcal{X}}})$. Thus, F-GP-UCB converges for $\mathcal{X}$, $k$ and $b$ such that the cumulative regret under standard GP-UCB without failure points is sublinear and $s(T) = \Theta(T)$.

Theorem 4.2 can be obtained from Theorem A.6 by letting $\mathcal{X} = [0, 1]^d, b(t) = 1/t^\alpha$ and using the infinity norm. We prove Lemma 4.1 before proving Theorem 4.2.

*Proof of lemma 4.1.* Using Lemma A.7 and Lemma A.3, we obtain

$$n_t \geq t - \mathcal{P}(F_c, \|\cdot\|_\infty, \theta_t b(t)) \tag{39}$$

$$\geq t - \mathcal{C}\left(F_c, \|\cdot\|_\infty, \frac{\theta_t b(t)}{2}\right) \tag{40}$$

$$\geq t - \mathcal{C}\left(\mathcal{X}, \|\cdot\|_\infty, \frac{\theta_t b(t)}{4}\right) \tag{41}$$

$$\geq t - \mathcal{P}\left(\mathcal{X}, \|\cdot\|_\infty, \frac{\theta_t b(t)}{4}\right). \tag{42}$$

$$\square$$

The following Lemma A.11 also holds.

**Lemma A.11.** *The following holds for any $t \in \mathbb{N}_+$:*

$$n_t \geq t - \mathcal{P}\left(\mathcal{X}, \|\cdot\|_\infty, \frac{\overline{\theta}b(t)}{4}\right). \tag{43}$$

We omit the proof of Lemma A.11 as it is similar to the proof of Theorem A.10. We give below the proof of Theorem 4.2.

*Proof of Theorem 4.2.* We consider the bounds for $\overline{\theta}$, $T_*$, and $s(T)$ under Lemma A.6 by letting $\mathcal{X} = [0, 1]^d$, $\|\cdot\| = \|\cdot\|_\infty$, $b(t) = 1/t^\alpha$.

- From $b(1) = 1$, $\overline{\theta} = \min\{\theta_0, \eta/(2b(1))\} = \min\{\theta_0, \eta/2\}$.

- From the definition of $b$, we can obtain $T_* = \lfloor (\theta_0/\eta)^{1/\alpha} + 1 \rfloor$ consistently as follows: We see that $b^{-1}(\cdot) = (\cdot)^{-1/\alpha}$ is also monotonically decreasing. Therefore, if $\eta/\theta_0 > b(1)$, then $b^{-1}(\eta/\theta_0) \in (0, 1)$ and $\lfloor b^{-1}(\theta_0/\eta) + 1 \rfloor = 1$, by which that $T_* = \lfloor (\theta_0/\eta)^{1/\alpha} + 1 \rfloor$ always holds is shown. Consequently, we obtain the upper bound of $T_*$: $T_* = \lfloor (\theta_0/\eta)^{1/\alpha} + 1 \rfloor \leq (\theta_0/\eta)^{1/\alpha} + 1$.

- Similarly to the proof of Lemma 4.1, we have $s(T) \geq T - \mathcal{P}\left(\mathcal{X}, \|\cdot\|_\infty, \overline{\theta}b(T)/4\right) = T - \lceil 4T^\alpha/\overline{\theta} \rceil^d$, where we use the fact that $\mathcal{P}([0, 1]^d, \|\cdot\|_\infty, \epsilon) = \lceil 1/\epsilon \rceil^d$ shown in Lemma A.4.

Then, we consider $T_s$. From the above derivation for $s(T)$, we can obtain $s(T) \geq T - \lceil 4T^\alpha/\overline{\theta} \rceil^d \geq T/2$ if $T \geq (10/\overline{\theta}^d)^{1/(1-d\alpha)}$ holds, as follows:

$$T \geq (10/\overline{\theta}^d)^{1/(1-d\alpha)} \Leftrightarrow T \geq \frac{10T^{d\alpha}}{\overline{\theta}^d} \Leftrightarrow \frac{T}{2} - \frac{5T^{d\alpha}}{\overline{\theta}^d} \geq 0, \tag{44}$$

from which we obtain

$$\frac{T}{2} \leq T - \left(\frac{5T^\alpha}{\overline{\theta}}\right)^d \tag{45}$$

$$= T - \left(\frac{4T^\alpha}{\overline{\theta}} + \frac{T^\alpha}{\overline{\theta}}\right)^d \tag{46}$$

$$\leq T - \left(\frac{4T^\alpha}{\overline{\theta}} + 2T^\alpha\right)^d \tag{47}$$

$$\leq T - \left(\frac{4T^\alpha}{\overline{\theta}} + 1\right)^d \tag{48}$$

$$\leq T - \left\lceil\frac{4T^\alpha}{\overline{\theta}}\right\rceil^d. \tag{49}$$

Note that $\overline{\theta} \leq \eta/2 \leq 1/2$, where $\eta \in (0, 1)$ from the definition of $\mathcal{X}$ and $\|\cdot\|_\infty$. Hence, $s(T) \geq 1$ for all $T \geq \max\{(10/\overline{\theta}^d)^{1/(1-d\alpha)}, 2\}$, which suffices to show $T_s \leq \tilde{T}_s := \max\{(10/\overline{\theta}^d)^{1/(1-d\alpha)}, 2\} + 1$.

Substituting the bounds on $\overline{\theta}, T_*, s(T), T_s$ obtained above into Theorem A.6 completes the proof. $\square$

# B   More general regret bound

A disadvantage of Theorem 4.2 is that the case where $x^*$ exists on the boundary of $S_c$ is not covered, which is the case that often appears in a real-world application. In this section, we give a more general regret bound that covers the case where $x^*$ on the boundary of $S_c$ and discuss its limitations.

We first assume that the failure function $c$ satisfies the following Assumption B.1 instead of Assumption 2.2.

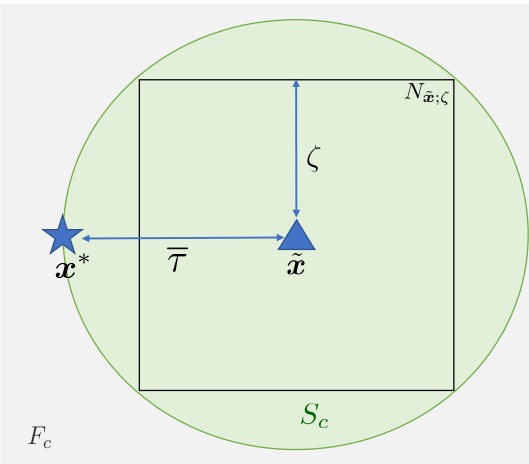

Figure 4: A two-dimensional example illustrating $\zeta$ and $\overline{\tau}$. In this example, the success region is a disc with $\boldsymbol{x}^*$ existing on its boundary. The point $\tilde{\boldsymbol{x}}$ is contained in the search space $\mathcal{X}_t$ at least at step $t$ such that $\theta_t b(t) < \zeta$. In this case, we have $f(\tilde{\boldsymbol{x}}) - f(\boldsymbol{x}^*) \leq L\overline{\tau}$ due to Lipschitz continuity.

**Assumption B.1.** *There exists a convex set $D \subset S_c$ satisfying the following conditions:*

    *1. The optimal solution $\boldsymbol{x}^*$ is included in $D$, namely, $\boldsymbol{x}^* \in D$.*

    *2. For some $\zeta > 0$, there exists $\tilde{\boldsymbol{x}} \in D$ such that $N_{\tilde{\boldsymbol{x}};\zeta} \subset D$.*

*Here, $N_{\tilde{\boldsymbol{x}};\zeta} := \{\boldsymbol{x} \in \mathcal{X} \mid \|\boldsymbol{x} - \tilde{\boldsymbol{x}}\|_\infty < \zeta\}$ denotes an open ball with a radius $\zeta$ centered at $\tilde{\boldsymbol{x}}$.*

Note that, for the failure function $c$ that satisfies Assumption 2.2, Assumption B.1 also holds for $c$ by letting $\tilde{\boldsymbol{x}} = \boldsymbol{x}^*$ and $\zeta = \eta$.

**Theorem B.2** (General regret bound.)**.** *Let $\beta^{1/2}, b, f, \delta, \mathcal{X}_t$, and $\alpha$ be defined as in Theorem 4.2. Assume that $f$ is $L$-Lipschitz continuous, that is*

$$\forall \boldsymbol{x}, \tilde{\boldsymbol{x}} \in \mathcal{X}, \ |f(\boldsymbol{x}) - f(\tilde{\boldsymbol{x}})| \leq L\|\boldsymbol{x} - \tilde{\boldsymbol{x}}\|_\infty. \tag{50}$$

*In addition, suppose that the failure function $c$ satisfies Assumption B.1. When applying Algorithm 1 under the above conditions, the following two statements hold:*

    *1. Estimated solution $\hat{\boldsymbol{x}}_T$ defined in (8) exists for $T \geq \tilde{T}_s$.*

    *2. The following holds with probability at least $1 - \delta$:*

$$\forall T \geq \tilde{T}_s, \ r_T \leq \frac{2}{T}\left[2B(\tilde{T}_* - 1) + \sqrt{C_1 \beta_T T \gamma_{T;S_c}} + L\theta_0\overline{\tau}\frac{T^{1-\alpha}}{(1-\alpha)\zeta}\right]. \tag{51}$$

*Here, we set $\tilde{T}_s := \max\{(10/\overline{\theta}^d)^{1/(1-d\alpha)}, 2\} + 1$, $\tilde{T}_* := (\theta_0/\zeta)^{1/\alpha} + 1$, and $\overline{\theta} := \min\{\theta_0, \zeta/2\}$. Furthermore, $C_1$ and $\overline{\tau}$ are defined as $C_1 = 8/\ln(1 + \sigma^{-2})$ and $\overline{\tau} = \|\boldsymbol{x}^* - \tilde{\boldsymbol{x}}\|_\infty$, respectively. It should be noted that $\zeta$ and $\tilde{\boldsymbol{x}}$ are defined in Assumption B.1.*

Note that the additional Lipschitz continuity on $f$ in Theorem B.2 is often assumed in existing GP optimization literature [43, 44, 27, 31]. In addition, the Lipschitzness assumption holds for commonly used kernels such as the Gaussian kernel [28].

In Theorem B.2, $\tilde{T}_s$ and $\overline{\theta}$ can be interpreted similarly to Theorem 4.2, where $\eta$ is replaced by $\zeta$. The regret bound in Theorem B.2 has a newly appearing third term containing the Lipschitz constant. This additional term converges to zero in $\mathcal{O}(T^{-\alpha}(1-\alpha)^{-1})$ as $T$ increases. An illustrative example of $\zeta$ and $\overline{\tau}$ is in Fig.4. Intuitively, the analysis of Theorem B.2 is obtained by applying Theorem 4.2 for $\tilde{\boldsymbol{x}}$ instead of $\boldsymbol{x}^*$; and we bound the regret incurred from the difference between $\boldsymbol{x}^*$ and $\tilde{\boldsymbol{x}}$ by using the Lipschitz property.

**Limitation of Theorem B.2.** When the first and second terms in (51) converge to zero, Theorem B.2 provides the no-regret guarantees, that is, $r_T \to 0$ (as $T \to \infty$) with high probability. Indeed, the third term in (51) is $\mathcal{O}(T^{-\alpha}(1-\alpha)^{-1})$ and always converges to zero as $T \to \infty$. However, since $\alpha \in (0, d^{-1})$, this third term incurs at least the regret of $\mathcal{O}(T^{-1/d})$, which dominates the second term on which the benefit of the kernelized assumption appears. We conjecture that our F-GP-UCB algorithm cannot avoid this undesirable third term and leaves its necessity analysis as future work. Here, although the result of Theorem B.2 is limited as described above, the following points should be noted:

- To our knowledge, even in the standard GP-based optimization that handles black-box constraint functions, the case where the optimum exists on the boundary of a feasible region is not considered in theory [43, 44, 5].

- The dependence of the failure function $c$ against the regret bound provided by Theorem B.2 is different from that of Theorem 4.2. Especially, the case exists where the bound of Theorem B.2 is tighter than that of Theorem 4.2 when $T$ is small.

- Existing analysis of [2] does not consider the case where $\boldsymbol{x}^*$ exists on the boundary of the success region. Our result of Theorem B.2 is the first convergence guarantee for the case where $\boldsymbol{x}^*$ exists on the boundary.

Finally, even if the result of Theorem B.2 is limited, we believe that it is valuable to show the convergence guarantees.

## B.1  Proof of Theorem B.2

As in the proof of Theorem 4.2 in Appendix A, the generalized Theorem B.3 is proved before Theorem B.2.

**Theorem B.3.** *Let* $\beta_t^{1/2} = B + \sigma\sqrt{2(\gamma_{t-1;\mathcal{X}} + 1 + \ln(1/\delta))}$ *for a fixed* $f \in \mathcal{H}_k$ *such that* $\|f\|_{\mathcal{H}_k} \leq B$ *and* $\delta \in (0,1)$. *We further assume the following four conditions:*

1. *$f$ is $L$-Lipschitz continuous over $\mathcal{X}$. That is,* $\forall \boldsymbol{x}^{(1)}, \boldsymbol{x}^{(2)} \in \mathcal{X}$, $|f(\boldsymbol{x}^{(1)}) - f(\boldsymbol{x}^{(2)})| \leq L\|\boldsymbol{x}^{(1)} - \boldsymbol{x}^{(2)}\|$.

2. *There exists a convex set $D \subset S_c$ such that $\boldsymbol{x}^* \in D$.*

3. *There exists some $\tilde{\boldsymbol{x}} \in D$ and $\zeta > 0$ such that $N_{\tilde{\boldsymbol{x}};\zeta,\|\cdot\|} \subset D$.*

4. *For $\overline{\theta} = \min\{\theta_0, \zeta/(2b(1))\}$, $\lim_{T \to \infty} (T - \mathcal{P}(F_c, \|\cdot\|, \overline{\theta}b(T))) = \infty$ holds.*

*Then, letting $s(T) = T - \mathcal{P}(F_c, \|\cdot\|, \overline{\theta}b(T))$, the following holds under Assumption A.5 in Algorithm 1:*

- *$\hat{\boldsymbol{x}}_T$ exists for some $T \in \mathbb{N}_+$ such that $T \geq T_s$.*

- *The following equation holds with probability at least $1 - \delta$:*

$$\forall T \geq T_s, r_T \leq \frac{1}{s(T)} \left[ 2B(T_* - 1) + \sqrt{C_1 \beta_T T \gamma_{T;S_c}} + \zeta^{-1} L \theta_0 \overline{\tau} \sum_{t=1}^{T} b(t) \right]. \quad (52)$$

*Here, $T_*$ is a natural number which is defined as follows:*

$$T_* = \begin{cases} \lfloor b^{-1}(\frac{\zeta}{\theta_0}) + 1 \rfloor & \text{if } \frac{\zeta}{\theta_0} \leq b(1), \\ 1 & \text{otherwise} \end{cases}. \quad (53)$$

*Here we let $T_s$ be the smallest natural number $T$ such that $s(T) \geq 1$, $\overline{\tau} = \|\boldsymbol{x}^* - \tilde{\boldsymbol{x}}\|$, and $C_1 = 8/\ln(1 + \sigma^{-2})$.*

The interpretation of $\overline{\theta}$, $T_s$, and $s(T)$ is the same as that in Theorem A.6. On the other hand, $T_*$ is the upper bound on the number of steps until $\mathcal{X}_t$ contains $\tilde{\boldsymbol{x}}$, the point "close" to $\boldsymbol{x}^*$. We give an illustrative example for $\tilde{\boldsymbol{x}}$, $\overline{\tau}$, and $\zeta$ in Fig.4.

The following Lemma B.4 evaluates the "closeness" of the points inside the convex region $D$ and $\boldsymbol{x}_*$. From this, $f(\boldsymbol{x}_t^*)$ and $f(\boldsymbol{x}^*)$ can be evaluated using Lipschitz continuity even if $\boldsymbol{x}^*$ does not exist inside $S_c$.

**Lemma B.4.** *For any $\tau \in (0, \overline{\tau}]$, where $\overline{\tau} = \|\tilde{\boldsymbol{x}} - \boldsymbol{x}^*\| > 0$, some $\boldsymbol{x}^{(\tau)} \in D$ exists such that the following holds:*

$$\|\boldsymbol{x}^{(\tau)} - \boldsymbol{x}^*\| = \tau \ \text{ and } \ N_{\boldsymbol{x}^{(\tau)};\zeta\tau/\overline{\tau},\|\cdot\|} \subset D. \tag{54}$$

*Proof.* Fix $\tau \in (0, \overline{\tau}]$. Let $\boldsymbol{x}^{(\tau)} = (1 - \tau/\overline{\tau})\boldsymbol{x}^* + (\tau/\overline{\tau})\tilde{\boldsymbol{x}}$. Then, we have $\|\boldsymbol{x}^* - \boldsymbol{x}^{(\tau)}\| = \tau$.

Let $\boldsymbol{x}_y = \boldsymbol{x}^* + (\boldsymbol{y} - \boldsymbol{x}^*)\overline{\tau}/\tau$ for an arbitrary $\boldsymbol{y} \in N_{\boldsymbol{x}^{(\tau)};\zeta\tau/\overline{\tau},\|\cdot\|}$. Then,

$$\|\boldsymbol{x}_y - \tilde{\boldsymbol{x}}\| = \|\boldsymbol{x}^* - \tilde{\boldsymbol{x}} + (\boldsymbol{y} - \boldsymbol{x}^*)\frac{\overline{\tau}}{\tau}\| \tag{55}$$

$$= \frac{\overline{\tau}}{\tau}\|\boldsymbol{y} - \left[\left(1 - \frac{\tau}{\overline{\tau}}\right)\boldsymbol{x}^* + \frac{\tau}{\overline{\tau}}\tilde{\boldsymbol{x}}\right]\| \tag{56}$$

$$= \frac{\overline{\tau}}{\tau}\|\boldsymbol{y} - \boldsymbol{x}^{(\tau)}\| \tag{57}$$

$$\leq \zeta. \tag{58}$$

It follows that $\boldsymbol{x}_y \in N_{\tilde{\boldsymbol{x}};\zeta,\|\cdot\|} \subset D$. Because $\boldsymbol{y} = (1 - \tau/\overline{\tau})\boldsymbol{x}^* + (\tau/\overline{\tau})\boldsymbol{x}_y$, we have $\boldsymbol{y} \in D$ from the convexity of $D$. This completes the proof. $\qquad\square$

Using Lemma B.4, we introduce another lemma which is similar to Lemma A.8.

**Lemma B.5.** *The following statements hold for any natural number $t$ such that $t \geq T_*$:*

1. *$\theta_t = \theta_{T_*}$ and $\theta_{T_*} \geq \min\{\theta_0, \zeta/(2b(1))\}$.*

2. *For any $\epsilon \in (0, \overline{\epsilon}]$, $\mathcal{X}_t$ contains $\boldsymbol{x}^{(\tau_t)}$ such that $\|\boldsymbol{x}^* - \boldsymbol{x}^{(\tau_t)}\| = \tau_t$, where $\tau_t = \theta_{T_*}b(t)\overline{\tau}/\zeta + \epsilon$. Here, we define $\overline{\epsilon}$ as $\overline{\epsilon} := \overline{\tau}(1 - \theta_{T_*}b(t)/\zeta)$.*

*Proof.* Statement 1 immediately follows by substituting $\zeta$ for $\eta$ in the proof of statement 2 of Lemma A.8:

Concerning statement 2, we have

$$\tau_t = \frac{\theta_{T_*}b(t)\overline{\tau}}{\zeta} + \epsilon \leq \frac{\theta_{T_*}b(t)\overline{\tau}}{\zeta} + \overline{\epsilon} \leq \overline{\tau}. \tag{59}$$

Thus, from Lemma B.4, there exists some $\boldsymbol{x}^{(\tau_t)}$ such that

$$\|\boldsymbol{x}^{(\tau_t)} - \boldsymbol{x}^*\| = \tau_t \ \text{ and } \ N_{\boldsymbol{x}^{(\tau_t)};\zeta\tau_t/\overline{\tau},\|\cdot\|} \subset D \subset S_c. \tag{60}$$

Moreover, from the definition of $\tau_t$, it follows that $\zeta\tau_t/\overline{\tau} > \theta_{T_*}b(t)$. Combining it with $N_{\boldsymbol{x}^{(\tau_t)};\zeta\tau_t/\overline{\tau},\|\cdot\|} \subset S_c$, we see that $N_{\boldsymbol{x}^{(\tau_t)};\theta_{T_*}b(t),\|\cdot\|} \subset S_c$, which implies $\|\boldsymbol{x}^{(\tau_t)} - \boldsymbol{x}\| > \theta_{T_*}b(t)$ for all $\boldsymbol{x} \in F_c$. Hence, we obtain $\boldsymbol{x}^{(\tau_t)} \in \mathcal{X}_t$. $\qquad\square$

We show below the proof of Theorem B.3.

*Proof of Theorem B.3.* The existence of $\hat{\boldsymbol{x}}_T$ follows immediately from Lemmas A.10 and the definition of $T_s$. Now, fix some natural number $T \geq T_s$ and assume Eq. (5) holds. Then, similarly to the proof of Theorem A.6, the following statements hold:

$$r_T \leq s(T)^{-1}\left[\sum_{t \in \mathcal{I}_T^{(S)}}(f(\boldsymbol{x}^*) - f(\boldsymbol{x}_t^*)) + \sum_{t \in \mathcal{I}_T^{(S)}}(f(\boldsymbol{x}_t^*) - f(\hat{\boldsymbol{x}}_T))\right]. \tag{61}$$

For the second term above, following the proof of Theorem A.6, we have

$$\sum_{t \in \mathcal{I}_T^{(S)}}(f(\boldsymbol{x}_t^*) - f(\hat{\boldsymbol{x}}_T)) \leq \sqrt{C_1\beta_T T\gamma_{T;S_c}}. \tag{62}$$

For the first term, we have

$$\sum_{t \in \mathcal{I}_T^{(S)}} (f(\boldsymbol{x}^*) - f(\boldsymbol{x}_t^*)) \leq \sum_{t=1}^{T} (f(\boldsymbol{x}^*) - f(\boldsymbol{x}_t^*)) \tag{63}$$

$$\leq \sum_{t=1}^{T_*-1} (f(\boldsymbol{x}^*) - f(\boldsymbol{x}_t^*)) + \sum_{t=T_*}^{T} (f(\boldsymbol{x}^*) - f(\boldsymbol{x}_t^*)) \tag{64}$$

$$\leq 2B(T_* - 1) + \sum_{t=T_*}^{T} (f(\boldsymbol{x}^*) - f(\boldsymbol{x}_t^*)). \tag{65}$$

From Lemma B.5, by using a sufficiently small $\epsilon$ and $\tau_t = \theta_{T_*} b(t) \overline{\tau} / \zeta + \epsilon$, we have

$$\sum_{t=T_*}^{T} (f(\boldsymbol{x}^*) - f(\boldsymbol{x}_t^*)) = \sum_{t=T_*}^{T} \left( f(\boldsymbol{x}^*) - f(\boldsymbol{x}^{(\tau_t)}) + f(\boldsymbol{x}^{(\tau_t)}) - f(\boldsymbol{x}_t^*) \right) \tag{66}$$

$$\leq \sum_{t=T_*}^{T} \left( f(\boldsymbol{x}^*) - f(\boldsymbol{x}^{(\tau_t)}) \right) \tag{67}$$

$$\leq \sum_{t=T_*}^{T} L \|\boldsymbol{x}^* - \boldsymbol{x}^{(\tau_t)}\| \tag{68}$$

$$= \sum_{t=T_*}^{T} L \tau_t \tag{69}$$

$$= L \theta_{T_*} \overline{\tau} \zeta^{-1} \left( \theta_{T_*}^{-1} \overline{\tau}^{-1} \zeta \epsilon + \sum_{t=T_*}^{T} b(t) \right) \tag{70}$$

$$\leq L \theta_0 \overline{\tau} \zeta^{-1} \left( \theta_{T_*}^{-1} \overline{\tau}^{-1} \zeta \epsilon + \sum_{t=1}^{T} b(t) \right). \tag{71}$$

The second line follows from the definition of $\boldsymbol{x}^{(\tau_t)} \in \mathcal{X}_t$ and $\boldsymbol{x}_t^*$. The third line follows from the Lipschitz continuity of $f$. The fourth line follows from Lemma B.5. In the equation above, letting $\epsilon \downarrow 0$ gives

$$\sum_{t=T_*}^{T} (f(\boldsymbol{x}^*) - f(\boldsymbol{x}_t^*)) \leq L \theta_0 \overline{\tau} \zeta^{-1} \sum_{t=1}^{T} b(t). \tag{72}$$

In summary, assuming Eq. (5) holds, we obtain

$$\forall T \geq T_s, \ r_T \leq \frac{1}{s(T)} \left[ 2B(T_* - 1) + \sqrt{C_1 \beta_T T \gamma_{T;S_c}} + \zeta^{-1} L \theta_0 \overline{\tau} \sum_{t=1}^{T} b(t) \right]. \tag{73}$$

Finally, by noting that Eq. (5) holds with probability at least $1 - \delta$, the proof is completed. $\qquad \square$

We show Theorem B.2 using Theorem B.3.

*Proof of theorem B.2.* We consider Theorem B.3 by letting $\mathcal{X} = [0,1]^d$, $\|\cdot\| = \|\cdot\|_\infty$, and $b(t) = 1/t^\alpha$. Now, by substituting $\zeta$ for $\eta$ in Theorem 4.2, we similarly have the following for $\overline{\theta}, T_*$, $T^*$, and $s(T)$.

- $\overline{\theta} = \min\{\theta_0, \eta/2\}$.

- $T_* \leq \tilde{T}_* := (\theta_0/\zeta)^{1/\alpha} + 1$.

- $T_s \leq \tilde{T}_s := \max\{(10/\overline{\theta}^d)^{1/(1-d\alpha)}, 2\} + 1$.

- For $T \geq \tilde{T}_s$, $s(T) \geq T/2$.

Using the inequality above and from Theorem B.3, we have

$$\forall T \geq \tilde{T}_s, \ r_T \leq \frac{2}{T} \left[ 2B(\tilde{T}_* - 1) + \sqrt{C_1 \beta_T T \gamma_{T;S_c}} + \zeta^{-1} L \theta_0 \overline{\tau} \sum_{t=1}^{T} t^{-\alpha} \right]. \tag{74}$$

For the third term, we have

$$\sum_{t=1}^{T} t^{-\alpha} \leq \int_0^T t^{-\alpha} \mathrm{d}t \leq \frac{T^{1-\alpha}}{1-\alpha}. \tag{75}$$

Substituting the equation above into Eq. (74) completes the proof.

$\square$

## C  Cumulative regret bound

In this section, we show the upper bound of the cumulative regret of Algorithm 1. We define the cumulative regret $R_T$ as

$$R_T = \sum_{t=1}^{T} \tilde{r}_t, \tag{76}$$

where $\tilde{r}_t$ is the instantaneous regret at step $t$, which is defined as

$$\tilde{r}_t = \begin{cases} f(\boldsymbol{x}^*) - f(\boldsymbol{x}_t) & \text{if } \boldsymbol{x}_t \in S_c, \\ f(\boldsymbol{x}^*) - \min_{\boldsymbol{x} \in \mathcal{X}} f(\boldsymbol{x}) & \text{if } \boldsymbol{x}_t \in F_c \end{cases}. \tag{77}$$

We show the following Theorem C.1.

**Theorem C.1.** *Fix $\delta \in (0,1)$ and $f \in \mathcal{H}_k$ such that $\|f\|_{\mathcal{H}_k} \leq B$. Let $\beta_t^{1/2} = B + \sigma\sqrt{2(\gamma_{t-1;\mathcal{X}} + 1 + \ln(1/\delta))}$. We further assume the following two conditions.*

1. *There exists $\eta > 0$ such that $N_{\boldsymbol{x}^*;\eta,\|\cdot\|} \subset S_c$ holds where $N_{\boldsymbol{x}^*;\eta,\|\cdot\|} := \{\boldsymbol{x} \in \mathcal{X} \mid \|\boldsymbol{x}-\boldsymbol{x}^*\| < \eta\}$.*

2. *$\lim_{T \to \infty} \left( T - \mathcal{P}\left( F_c, \|\cdot\|, \overline{\theta}b(T) \right) \right) = \infty$ holds for $\overline{\theta} = \min\{\theta_0, \eta/(2b(1))\}$.*

*Then, under Assumption A.5, the following holds with probability at least $1 - \delta$ in Algorithm 1:*

$$\forall T \geq 1, R_T \leq 2B \left( \mathcal{P}\left( F_c, \|\cdot\|, \overline{\theta}b(T) \right) + T_* - 1 \right) + \sqrt{C_1 \beta_T T \gamma_{T;S_c}}. \tag{78}$$

*Here, $T_*$ is a natural number defined as follows:*

$$T_* = \begin{cases} \lfloor b^{-1}(\frac{\eta}{\theta_0}) + 1 \rfloor & \text{if } \frac{\eta}{\theta_0} \leq b(1), \\ 1 & \text{otherwise} \end{cases}. \tag{79}$$

*Moreover, we set $C_1$ as $C_1 = 8/\ln(1 + \sigma^{-2})$.*

*Proof.* Under the event (5), the following holds:

$$R_T = \sum_{t \in \mathcal{I}_T^{(F)}} \tilde{r}_t + \sum_{t \in \mathcal{I}_T^{(S)}} \tilde{r}_t$$

$$\leq 2B|\mathcal{I}_T^{(F)}| + \sum_{t \in \mathcal{I}_T^{(S)}} (f(\boldsymbol{x}^*) - f(\boldsymbol{x}_t)) \tag{80}$$

$$\leq 2B|\mathcal{I}_T^{(F)}| + \sum_{t \in \mathcal{I}_T^{(S)}} (f(\boldsymbol{x}^*) - f(\boldsymbol{x}_t^*)) + \sum_{t \in \mathcal{I}_T^{(S)}} (f(\boldsymbol{x}_t^*) - f(\boldsymbol{x}_t))$$

$$\leq 2B|\mathcal{I}_T^{(F)}| + \sum_{t=1}^{T_*-1} (f(\boldsymbol{x}^*) - f(\boldsymbol{x}_t^*)) + \sum_{t \in \mathcal{I}_T^{(S)}} (\mathrm{ucb}_t(\boldsymbol{x}_t) - \mathrm{lcb}_t(\boldsymbol{x}_t)) \tag{81}$$

$$\leq 2B \left( |\mathcal{I}_T^{(F)}| + T_* - 1 \right) + 2\beta_T^{1/2} \sum_{t \in \mathcal{I}_T^{(S)}} \sigma_{t-1}(\boldsymbol{x}_t)$$

$$\leq 2B \left( \mathcal{P}\left(F_c, \|\cdot\|, \bar{\theta}b(T)\right) + T_* - 1 \right) + \sqrt{C_1 \beta_T T \gamma_{T;S_c}}, \tag{82}$$

where:

- In (80), the first term follows by applying the inequality $\tilde{r}_t \leq 2B$, and the second term follows by using the fact that $\tilde{r}_t = f(\boldsymbol{x}^*) - f(\boldsymbol{x}_t)$ holds for $\boldsymbol{x}_t \in S_c$.

- In (81), the second term follows from Lemma A.8, which shows $\boldsymbol{x}^* \in \mathcal{X}_t$ for any $t \geq T_*$. The third term follows by using (5).

- In (82), the first term follows by applying the inequality $|\mathcal{I}_T^{(F)}| \leq \mathcal{P}\left(F_c, \|\cdot\|, \bar{\theta}b(T)\right)$ from Lemma A.10. The second term follows by using Lemma A.9 and the monotonicity of the maximum information gain.

Finally, the event (5) holds with probability at least $1 - \delta$. This implies that (78) holds with probability at least $1 - \delta$. □

Furthermore, as in Theorem 4.2, the following Corollary C.2 gives the result for the case where $\mathcal{X}$, $b(t)$, and the norm $\|\cdot\|$ are defined as $\mathcal{X} = [0,1]^d$, $b(t) = 1/t^\alpha$ with $\alpha \in (0, d^{-1})$, and $\|\cdot\| = \|\cdot\|_\infty$, respectively.

**Corollary C.2.** *Fix $\delta \in (0,1)$, $\alpha \in (0, d^{-1})$, $\mathcal{X} = [0,1]^d$ and $f \in \mathcal{H}_k$ such that $\|f\|_{\mathcal{H}_k} \leq B$. Let $\beta_t^{1/2} = B + \sigma\sqrt{2(\gamma_{t-1;\mathcal{X}} + 1 + \ln(1/\delta))}$ and $b(t) = t^{-\alpha}$. In addition, there exists $\eta > 0$ such that $N_{\boldsymbol{x}^*;\eta} \subset S_c$, where $N_{\boldsymbol{x}^*;\eta} = \{\boldsymbol{x} \in \mathcal{X} \mid \|\boldsymbol{x} - \boldsymbol{x}^*\|_\infty < \eta\}$. When applying Algorithm 1 under the above conditions, with probability at least $1 - \delta$, the following upper bound of the cumulative regret holds:*

$$\forall T \geq 1, R_T \leq 2B \left[ \left( 5\bar{\theta}^{-1} \right)^d T^{\alpha d} + \tilde{T}_* - 1 \right] + \sqrt{C_1 \beta_T T \gamma_{T;S_c}}. \tag{83}$$

*Here, we set $\tilde{T}_*$, $\bar{\theta}$ and $C_1$ as $\tilde{T}_* := (\theta_0/\eta)^{1/\alpha} + 1$, $\bar{\theta} := \min\{\theta_0, \eta/2\}$, and $C_1 = 8/\ln(1 + \sigma^{-2})$, respectively.*

*Proof.* From Theorem C.1, it is sufficient to prove that $\mathcal{P}\left(F_c, \|\cdot\|, \bar{\theta}b(T)\right) \leq (5/\bar{\theta})^d T^{\alpha d}$ and $T_* \leq \tilde{T}^*$ holds, where $T_*$ is defined in Theorem C.1. Under $\mathcal{X} = [0,1]^d$ with the norm $\|\cdot\|_\infty$ and $b(t) = t^{-\alpha}$, the inequality $\mathcal{P}\left(F_c, \|\cdot\|, \bar{\theta}b(T)\right) \leq \mathcal{P}\left(\mathcal{X}, \|\cdot\|, \bar{\theta}b(T)/4\right) \leq \lceil 4T^\alpha/\bar{\theta} \rceil^d$ holds from Lemma A.11 and Lemma A.4. Furthermore, the inequality $\lceil 4T^\alpha/\bar{\theta} \rceil^d \leq (5/\bar{\theta})^d T^{\alpha d}$ follows from the fact of $\bar{\theta} \leq 1$ and $T \geq 1$. Consequently, $\mathcal{P}\left(F_c, \|\cdot\|, \bar{\theta}b(T)\right) \leq (5/\bar{\theta})^d T^{\alpha d}$ holds. As for $T_*$, we already obtained the inequality $T_* \leq \tilde{T}^*$ in the proof of Theorem 4.2. This completes the proof. □

# D The difficulty to extend the algorithm of [48] and [30]

As described in Sec. 4, the regret bound of the F-GP-UCB algorithm has the $\tilde{O}(\gamma_{T;\mathcal{X}}/\sqrt{T})$ term, where the notation $\tilde{\mathcal{O}}(\cdot)$ denotes the order whose dimension-independent logarithmic factors are hidden. Similarly, as described in Appendix C, the cumulative regret bound of the F-GP-UCB algorithm has the $O(\gamma_{T;\mathcal{X}}\sqrt{T})$ term. Although whether the GP-UCB based algorithm achieves $\tilde{O}(\gamma_{T;\mathcal{X}}/\sqrt{T})$ regret or not is an open problem even in the standard setting [50], several existing works proposed algorithms which achieve the regret of $\tilde{O}(\sqrt{\gamma_{T;\mathcal{X}}/T})$, or the cumulative regret of $\tilde{O}(\sqrt{\gamma_{T;\mathcal{X}}T})$ [51, 48, 40, 7, 30].

In this section, we discuss some naive extensions of the algorithm of [48] and [30] to our settings and clarify the difficulty we encountered while building their theoretical analysis in our setup. As a summary, due to the existence of the unknown failure regions, we could not naively extend the algorithm of [48] and [30] to our setup. We leave the study of the algorithm that achieves the regret of $\tilde{\mathcal{O}}(\sqrt{\gamma_{T;\mathcal{X}}/T})$ as future work.

## D.1 Maximum variance reduction based algorithm of [48]

**Algorithm overview**   The maximum variance reduction (MVR) algorithm in the standard setting defines the observation point $\boldsymbol{x}_t$ at each step $t$ as $\boldsymbol{x}_t = \operatorname{argmax}_{\boldsymbol{x} \in \mathcal{X}} \sigma_{t-1}(\boldsymbol{x})$, and define the estimated solution $\hat{\boldsymbol{x}}_t$ as $\hat{\boldsymbol{x}}_t = \operatorname{argmax}_{\boldsymbol{x} \in \mathcal{X}} \mu_t(\boldsymbol{x})$. From Theorem 3 of [48], the MVR algorithm achieves the $\tilde{\mathcal{O}}(\sqrt{\gamma_{T;\mathcal{X}}/T})$ regret with high probability. The core idea of their analysis is that the tight confidence bound of $f$ can be obtained when the selection rule of $\boldsymbol{x}_t$ is *non-adaptive*, which means $\{\boldsymbol{x}_t\}$ and the noises $\{\epsilon_t\}$ are independent. Since the $\{\boldsymbol{x}_t\}$ of MVR is chosen only from the posterior variance, which is independent of the noises, the non-adaptive tight confidence bound of $f$ can be obtained as in Theorem 1 of [48]. We give a brief summary of their regret analysis below:

1. From the definition of $\boldsymbol{x}_t$ of MVR, the maximum posterior variance is monotonically decreasing, i.e., $\tilde{t} \le t \Rightarrow \sigma_{\tilde{t}-1}(\boldsymbol{x}_{\tilde{t}}) \ge \sigma_{t-1}(\boldsymbol{x}_t)$. By combining the monotonicity of $\sigma_{t-1}(\boldsymbol{x}_t)$ with the inequality of maximum information gain, the posterior standard deviation of any input at step $T$ is bounded by $\mathcal{O}(\sqrt{\gamma_{T;\mathcal{X}}/T})$. More specifically, the following holds:

$$\forall \boldsymbol{x} \in \mathcal{X}, \sigma_T(\boldsymbol{x}) \le \sqrt{\frac{2\gamma_{T;\mathcal{X}}}{\ln(1+\sigma^{-2})T}}. \tag{84}$$

2. The regret is decomposed by using the posterior mean of $\hat{\boldsymbol{x}}_T$ as follows:

$$\begin{aligned} r_T &= f(\boldsymbol{x}^*) - f(\hat{\boldsymbol{x}}_T) \\ &= \underbrace{f(\boldsymbol{x}^*) - \mu_T(\boldsymbol{x}^*)}_{(a)} + \underbrace{\mu_T(\hat{\boldsymbol{x}}_T) - f(\hat{\boldsymbol{x}}_T)}_{(b)}. \end{aligned} \tag{85}$$

Here, (85) follows from the fact: $\forall \boldsymbol{x} \in \mathcal{X}, \mu_T(\hat{\boldsymbol{x}}_T) \ge \mu_T(\boldsymbol{x})$.

3. With probability $1 - \delta/3$, the upper bound of the term (a) is obtained as $\beta\sigma_T(\boldsymbol{x}^*)$, where $\beta$ is the width of confidence bound of $f(\boldsymbol{x}^*)$ which is the constant for fixed $\delta$, and does not depend on $T$. By combining (84), we obtain the upper bound of $\mathcal{O}(\sqrt{\gamma_{T;\mathcal{X}}/T})$ of the term (a).

4. Under the regularity condition for the kernel $k$ as in Assumption 4 in [48], with probability $1 - 2\delta/3$, the upper bound of $\tilde{\mathcal{O}}(\sqrt{\gamma_{T;\mathcal{X}}/T} + 1/\sqrt{T})$ of the term (b) is obtained.

5. By taking the union bound of step 3 and 4, the regret upper bound of $\tilde{\mathcal{O}}(\sqrt{\gamma_{T;\mathcal{X}}/T})$ is obtained with high probability.

**The naive extension to our setting.**   The naive extension of MVR to our setting can be considered by modifying the search space at step $t$ of the MVR algorithm from $\mathcal{X}$ to $\mathcal{X}_t$. More specifically, in the modified MVR algorithm, the observation point $\boldsymbol{x}_t$ and the estimated solution $\hat{\boldsymbol{x}}_t$ at step $t$ are respectively defined as follows:

$$\boldsymbol{x}_t = \operatorname{argmax}_{\boldsymbol{x} \in \mathcal{X}_t} \sigma_{t-1}(\boldsymbol{x}), \tag{86}$$

$$\hat{\boldsymbol{x}}_t = \operatorname{argmax}_{\boldsymbol{x} \in \mathcal{X}_t} \mu_t(\boldsymbol{x}). \tag{87}$$

Here, $\mathcal{X}_t$ is defined as

$$\mathcal{X}_t = \{\boldsymbol{x} \in \mathcal{X} \mid \forall i \in \mathcal{I}_{t-1}^{(F)}, \|\boldsymbol{x} - \boldsymbol{x}_i\|_\infty \geq \theta_t b_T\}, \tag{88}$$

where $\theta_t$ is a parameter which is determined in the same way as F-GP-UCB, and $b_T$ is a parameter which depends on the total step size $T$. It should be noted here that we consider the setting where total step size $T$ is known as the same as the setting in [48], whereas our main paper considers the setting where total step size $T$ is unknown. Therefore, we consider the fixed parameter $b_T$ instead of the decreasing function $b(\cdot)$ to define the search space $\mathcal{X}_t$. The pseudo-code of this modified MVR algorithm is given by replacing the lines 7, 8, and 16 of $\mathcal{X}_t$, $\boldsymbol{x}_t$, and $\hat{\boldsymbol{x}}_t$ in the Algorithm 1 with (86), (87), and (88). Since the failure function $c$ is fixed, $\{\boldsymbol{x}_t\}$ and $\{\epsilon_t\}$ are independent in this modified algorithm, and we can leverage the tight confidence bound of $f$ as in Theorem 4 of [48]. Furthermore, by using the packing argument described in Sec.4, the number of successful observations can be secured with the properly pre-specified parameter $b_T$. However, in the modified MVR algorithm described above, there exist two crucial problems to building the theoretical guarantees as follows:

- **The monotonicity of the posterior variance is not guaranteed.** Since the modified MVR algorithm chooses $\boldsymbol{x}_t$ in the search space $\mathcal{X}_t$, which varies along the time horizon $t$, the monotonicity of the posterior variance $\sigma_{t-1}(\boldsymbol{x}_t)$ does not always holds even if we only focus on the time steps which the successful observations are obtained. This is because lines 4-6 in Algorithm 1 could expand the search space $\mathcal{X}_t$. If the user specify a small parameter $\theta_0$ such that $\theta_0 b_T < \eta$ holds, the monotonicity of $\sigma_{t-1}(\boldsymbol{x}_t)$ is guaranteed. However, whether the parameter $\theta_0$ can be pre-specified such that $\theta_0 b_T < \eta$, there requires unrealistic prior knowledge about $\eta$.

- **The estimated solution is not guaranteed to become feasible.** Since the presence of the failure point in the $\mathcal{X}_t$, we could not guarantee the estimated solution is feasible. The F-GP-UCB algorithm avoids this problem by defining the estimated solution within past successful inputs.

### D.2 Batched pure exploration based algorithm of [30]

The batched pure exploration (BPE) [30] is an algorithm that observes the points based on the maximum posterior variance within an appropriate batch size. In addition, the BPE algorithm reduces the search space (which is called potential maximizers) according to the maximum lower confidence bound at the end of each batch. Here, for simplicity, we consider the case where $\mathcal{X}$ is finite.

Algorithm 2 shows one possible extension of the BPE algorithm to our setup, in which the past failure points are eliminated from the potential maximizers. In Algorithm 2, we use the following notations:

- The posterior mean and variance at the end of the $j-1$ step for batch $i$ is defined as $\mu_{j-1}^i$ and $\sigma_{j-1}^i$, respectively.

- The upper and lower confidence bounds of $f(\boldsymbol{x})$ at the end of batch $i$ are denoted as $\mathrm{ucb}^i(\boldsymbol{x})$ and $\mathrm{lcb}^i(\boldsymbol{x})$, respectively. Namely, $\mathrm{ucb}^i(\boldsymbol{x}) = \mu_{\bar{j}}^i(\boldsymbol{x}) + \beta\sigma_{\bar{j}}^i(\boldsymbol{x})$ and $\mathrm{lcb}^i(\boldsymbol{x}) = \mu_{\bar{j}}^i(\boldsymbol{x}) - \beta\sigma_{\bar{j}}^i(\boldsymbol{x})$, where $\bar{j}$ is the last index within the batch and $\beta$ is the parameter which represents the width of the confidence bound.

Note that additional operations with respect to the original BPE algorithm are lines 10–12, which eliminates the failure points, and lines 14 and 20–22, which count and secure the number of successful observations in the batch. In addition, since the failure function $c$ is unknown to the user, the observation trials are done at each iteration within each batch (line 9); on the other hand, the original algorithm observes all $N_i$ suggested points $\boldsymbol{x}_t$ at the end of the batch all at once. Unfortunately, under a natural assumption that $T < |\mathcal{X}|$, we cannot exclude the possibility that the algorithm gets stuck in the first batch by failing the observation $T$ times.

Although other extensions may be able to secure enough number of successful observations, they appear to break other essential requirements to achieve the $\tilde{\mathcal{O}}(\sqrt{\gamma_{T;\mathcal{X}}T})$ cumulative regret. For example, we can consider an idea that is similar to F-GP-UCB, i.e., the exclusion of the neighborhood of the observation failure from the search space instead of line 11 in Algorithm 2, which can secure enough number of successful observations by appropriately reducing $\theta_t$. However, since the search

space can be increased by reducing $\theta_t$, the monotonicity of the variance at observed points cannot be guaranteed. This monotonicity is a key property for $\tilde{\mathcal{O}}(\sqrt{\gamma_{T;\mathcal{X}}T})$ cumulative regret of the BPE algorithm, as with the MVR algorithm. As with the discussion for the MVR algorithm, although using a pre-specified and sufficiently small $\theta_0$ can guarantee the monotonicity of the variance, setting $\theta_0$ beforehand requires unrealistic prior knowledge. Furthermore, if $\theta_0 b_T < \|\boldsymbol{x} - \boldsymbol{x}'\|$ for all $\boldsymbol{x}, \boldsymbol{x}' \in \mathcal{X}$ such that $\boldsymbol{x} \neq \boldsymbol{x}'$, then the exclusion of the neighborhood of the observation failure is equivalent to line 11 in Algorithm 2. Thus, the pre-specified sufficiently small $\theta_0$ can cause again the problem that the algorithm makes the failure observation $T$ times in the first batch as with Algorithm 2. In conclusion, we believe that the BPE-based algorithm for our setup is not obvious.

---

**Algorithm 2** The naive extension of the batched pure exploration-based algorithm.

---

**Input:** Total step size $T$, the discrete input space $\mathcal{X}$.
1: Initialize GP prior.
2: $\overline{\mathcal{X}}_1 \leftarrow \mathcal{X}, t \leftarrow 1, N_0 \leftarrow 1$.
3: **for** $i = 1, 2, \ldots$ **do**
4:     $N_i \leftarrow \sqrt{TN_{i-1}}$.
5:     $S_i \leftarrow \emptyset$.
6:     **for** $j = 1, 2, \ldots$ **do**
7:         Compute $\sigma_{j-1}^i$ only based on $S_i$.
8:         $\boldsymbol{x}_t \leftarrow \mathrm{argmax}_{\boldsymbol{x} \in \overline{\mathcal{X}}_i} \sigma_{j-1}^i(\boldsymbol{x})$.
9:         Try to observe at $\boldsymbol{x}_t$.
10:       **if** $c(\boldsymbol{x}_t) = 1$ **then**
11:         $\overline{\mathcal{X}}_i \leftarrow \overline{\mathcal{X}}_i \setminus \{\boldsymbol{x}_t\}$.
12:       **else**
13:         $S_i \leftarrow S_i \cup \{\boldsymbol{x}_t\}$.
14:         $m_i \leftarrow m_i + 1$.
15:       **end if**
16:       $t \leftarrow t + 1$.
17:       **if** $t > T$ **then**
18:         Terminate.
19:       **end if**
20:       **if** $m_i = N_i$ **then**
21:         **break**.
22:       **end if**
23:     **end for**
24:     Compute $\mu_j^i$ and $\sigma_j^i$ only based on $S_i$.
25:     $\overline{\mathcal{X}}_{i+1} \leftarrow \{\boldsymbol{x} \in \overline{\mathcal{X}}_i \mid \mathrm{ucb}^i(\boldsymbol{x}) \geq \max_{\tilde{\boldsymbol{x}} \in \overline{\mathcal{X}}_i} \mathrm{lcb}^i(\tilde{\boldsymbol{x}})\}$.
26: **end for**

---

# E   Details of Section 5

## E.1   Details of computation of $\theta_t$

In this subsection, we give the details of the proposed algorithm for computation of $\theta_t$ which is described briefly in Sec. 5. In the proposed algorithm, before running the solver that uses $\theta_{t-1}$, a decision based on the packing number $\mathcal{P}(\mathcal{X}, \|\cdot\|_\infty, \theta_{t-1}b(t))$ is made to exclude cases where it is clear that a feasible solution could not be obtained. In fact, if the failure number $|\mathcal{I}_{t-1}^{(F)}|$ satisfies $|\mathcal{I}_{t-1}^{(F)}| \geq \mathcal{P}(\mathcal{X}, \|\cdot\|_\infty, \theta_{t-1}b(t))$, then, by the definition of $\mathcal{P}$, there are no feasible solutions. Then, $\theta_{t-1}$ is multiplied by $1/2$ in the proposed algorithm. This is repeated until the condition $\mathcal{P}(\mathcal{X}, \|\cdot\|_\infty, \theta_{t-1}b(t)) = \lceil 1/\theta_{t-1}b(t) \rceil^d > |\mathcal{I}_{t-1}^{(F)}|$ is met. After these shrinking procedures of $\theta_{t-1}$, we try to solve the problem (7); and if a feasible solution could not be obtained, $\theta_{t-1}$ is multiplied by $1/2$ and problem (7) is solved again. The proposed procedure is shown in Algorithm 3, which is used in lines 3–8 of Algorithm 1.

Lemma E.1 below shows that a feasible solution $\boldsymbol{x}_t$ is guaranteed to be obtained within two optimization trials. The proof is shown in Appendix F.

---

**Algorithm 3** Practical method to compute $\theta_t$ and $\boldsymbol{x}_t$.

---

**Input:** $\theta_t, b(t)$.

1: $\tilde{\theta}_t \leftarrow \theta_{t-1}$.

2: **while** $\lceil 1/\tilde{\theta}_t b(t)\rceil^d \leq |\mathcal{I}_{t-1}^{(F)}|$ **do**

3:   $\tilde{\theta}_t \leftarrow \tilde{\theta}_t/2$.

4: **end while**

5: Solve problem $\boldsymbol{x}_t = \underset{\boldsymbol{x}\in\mathcal{X}}{\arg\max}\,\mathrm{ucb}_t(\boldsymbol{x})$ s.t. $\forall i \in \mathcal{I}_{t-1}^{(F)}, \|\boldsymbol{x}_i - \boldsymbol{x}\|_\infty \geq \tilde{\theta}_t b(t)$.

6: **if** solver did not return any feasible solution $\boldsymbol{x}_t$, **then**

7:   $\theta_t \leftarrow \tilde{\theta}_t/2$.

8:   $\boldsymbol{x}_t = \underset{\boldsymbol{x}\in\mathcal{X}}{\arg\max}\,\mathrm{ucb}_t(\boldsymbol{x})$ s.t. $\forall i \in \mathcal{I}_{t-1}^{(F)}, \|\boldsymbol{x}_i - \boldsymbol{x}\|_\infty \geq \theta_t b(t)$.

9: **else**

10:   $\theta_t \leftarrow \tilde{\theta}_t$.

11: **end if**

**Output:** $\theta_t, \boldsymbol{x}_t$.

---

**Lemma E.1.** *Let $\mathcal{I}_{t-1}^{(F)}$ be the index set of failure points obtained up to step $t-1$ for a fixed $t$. Then, for any $\theta > 0$ such that $\mathcal{P}(\mathcal{X}, \|\cdot\|_\infty, \theta b(t)) > |\mathcal{I}_{t-1}^{(F)}|$, the following holds.*

$$\left\{\boldsymbol{x} \in \mathcal{X} \mid \forall i \in \mathcal{I}_{t-1}^{(F)}, \|\boldsymbol{x}_i - \boldsymbol{x}\|_\infty \geq \frac{\theta b(t)}{2}\right\} \neq \emptyset. \tag{89}$$

### E.2 Discussion about selection of $\alpha$

The choice of $\alpha$ in the proposed algorithm affects the initial number of crash points and the guaranteed rate of regret convergence. If $\alpha$ approaches $1/d$, from Theorems 4.2 and B.2, then $\tilde{T}_s$, i.e., the number of steps needed for a successful observation, becomes large, leading to an enormous number of steps needed for converging behavior. Despite this, the convergence rate eventually becomes fast due to the rate improvement in $T^*$ and the third term in Theorem B.2. Thus, a rational choice is to pick the intermediate value $\alpha = 1/(2d)$ especially if there are no insights about the required total number of steps.

### E.3 Pseudo-code of adaptive tuning method of $\theta_0$

We show the variants of the proposed method in Algorithm 4, which consider the adaptive tuning of $\theta_0$ as described in the final paragraph of Sec. 5. Furthermore, the pseudo-code, which includes all practical considerations discussed in Sec. 5, is in Algorithm 5.

## F Proofs of Section 5 and Appendix E

We first give a proof of Lemma E.1.

*Proof of Lemma E.1.* It is sufficient to show $|\mathcal{I}_{t-1}^{(F)}| < \mathcal{C}(\mathcal{X}, \|\cdot\|_\infty, \theta b(t)/2)$. From Lemma A.3 and the definition of $\theta$,

$$|\mathcal{I}_{t-1}^{(F)}| < \mathcal{P}(\mathcal{X}, \|\cdot\|_\infty, \theta b(t)) \leq \mathcal{C}\left(\mathcal{X}, \|\cdot\|_\infty, \frac{\theta b(t)}{2}\right). \tag{90}$$

□

Next, the following Lemma F.1 gives the regret when letting $b(t) = 1/\ln(t+1)$ in Theorem B.3.

**Lemma F.1.** *Substituting $b$ with $b(t) = 1/\ln(t+1)$ under the conditions of Theorem B.2, the following holds when executing Algorithm 1.*

- *$\hat{\boldsymbol{x}}_T$ exists for some $T \in \mathbb{N}_+$ such that $T \geq \tilde{T}_s$.*

**Algorithm 4** The F-GP-UCB algorithm with adaptive tuning method of $\theta_0$.

**Input:** GP prior $\mathcal{GP}(0,\ k)$, $\theta_{\min}, \theta_{\max} \in (0,1)$, $h_\sigma > 0$, $q \in \mathbb{N}_+$, $w \in (0,1)$, $b : \mathbb{N} \to \mathbb{R}_+$, $\{\beta_t\}_{t\in\mathbb{N}_+}$.

1: Initialize $\mathcal{I}_0^{(S)} = \mathcal{I}_0^{(F)} = \emptyset$, $\theta_0 = \theta_{\max}$, $C = 0$.
2: **for** $t = 1$ to $T$ **do**
3:     $\tilde{\theta}_t \leftarrow \theta_{t-1}$.
4:     **while** $\mathcal{X} \subset \bigcup_{i\in\mathcal{I}_t^{(F)}} N_{\boldsymbol{x}_i;\tilde{\theta}_t b(t)}$ **do**
5:         $\tilde{\theta}_t \leftarrow \tilde{\theta}_t/2$.
6:     **end while**
7:     $\theta_t \leftarrow \tilde{\theta}_t$, $\mathcal{X}_t \leftarrow \{\boldsymbol{x} \in \mathcal{X} \mid \forall i \in \mathcal{I}_{t-1}^{(F)}, \|\boldsymbol{x}_i - \boldsymbol{x}\|_\infty \geq \theta_t b(t)\}$.
8:     Choose $\boldsymbol{x}_t = \mathrm{argmax}_{\boldsymbol{x}\in\mathcal{X}_t}\mathrm{ucb}_t(\boldsymbol{x})$.
9:     **if** $c(\boldsymbol{x}_t) = 0$ **then**
10:         Observe $y_t = f(\boldsymbol{x}_t) + \epsilon_t$.
11:         Update GP by adding $(\boldsymbol{x}_t, y_t)$.
12:         $\mathcal{I}_t^{(S)} \leftarrow \mathcal{I}_{t-1}^{(S)} \cup \{t\}, \mathcal{I}_t^{(F)} \leftarrow \mathcal{I}_{t-1}^{(F)}$.
13:     **else**
14:         $\mathcal{I}_t^{(S)} \leftarrow \mathcal{I}_{t-1}^{(S)}, \mathcal{I}_t^{(F)} \leftarrow \mathcal{I}_{t-1}^{(F)} \cup \{t\}$.
15:     **end if**
16:     **if** $\sigma_{t-1}(\boldsymbol{x}_t) < h_\sigma$ **then**
17:         $C \leftarrow C + 1$.
18:         **if** $C = q$ **then**
19:             $\theta_t \leftarrow w\theta_t$.
20:             $C \leftarrow 0$.
21:         **end if**
22:     **else**
23:         $C \leftarrow 0$.
24:     **end if**
25: **end for**
26: $\hat{t} = \mathrm{argmax}_{i\in\mathcal{I}_t^{(S)}}\mathrm{lcb}_i(\boldsymbol{x}_i)$.

**Output:** $\boldsymbol{x}_{\hat{t}}$.

- *The following equation holds with probability at least $1 - \delta$:*

$$\forall T \geq \tilde{T}_s, r_T \leq \frac{2}{T}\left[2B(\tilde{T}_* - 1) + \sqrt{C_1\beta_T T\gamma_{T;S_c}} + \zeta^{-1}L\theta_0\tau C_2\frac{T}{\ln T}\right]. \quad (91)$$

*Here, we let $\tilde{T}_* = \exp(\zeta/\eta)$, $\tilde{T}_s = \max\{(10/\overline{\theta}^d)^{1/(1-d\alpha)}, 2\}+1$, $\overline{\theta} = \min\{\theta_0, (\zeta\ln 2)/2\}$, and $C_1 = 8/\ln(1 + \sigma^{-2})$. Moreover, $C_2$ is an absolute constant.*

*Proof.* We evaluate $s(T), T_s, T_*, \overline{\theta}$, and $\sum_{t=1}^T b(t)$ when substituting $b(t) = 1/\ln(t+1)$ in Theorem B.3. $\overline{\theta}$ can be bounded from above by $\tilde{T}_s$ from $b(1) = 1/\ln 2$. $\overline{\theta} = \min\{\theta_0, \zeta\ln 2/2\}$, $T_s$ can be bounded from above by $\tilde{T}_s$ similarly to Theorem B.2 We similarly have $s(T) \geq T/2$ from $T \geq \tilde{T}_s$. Concerning $T_*$, we have $b^{-1}(\zeta/\theta_0) = \exp(\theta_0/\zeta)$ and $T_* \leq \exp(\theta_0/\zeta) := \tilde{T}_*$. Lastly, we have

$$\sum_{t=1}^T \frac{1}{\ln(t+1)} \leq \int_0^{T+1} \frac{1}{\ln(t+1)}\mathrm{d}t \quad (92)$$

$$\leq C_2\frac{T}{\ln T}. \quad (93)$$

Here, in the last line, we used the known fact that the order of logarithmic integral is $\mathcal{O}(T/\ln T)$. $C_2$ is an absolute constant given by the logarithmic integral. Applying the obtained results for $s(T), T_s, T_*, \overline{\theta}$, and $\sum_{t=1}^T b(t)$ in Theorem B.3 completes the proof. $\square$

Next, we prove Theorem F.2 and Theorem F.3, which we describe below.

**Algorithm 5** The practical version of F-GP-UCB algorithm.

**Input:** GP prior $\mathcal{GP}(0,\ k)$, $\theta_{\min}, \theta_{\max} \in (0,1)$, $h_\sigma > 0$, $q \in \mathbb{N}_+$, $w \in (0,1)$, $b : \mathbb{N} \to \mathbb{R}_+$, $\{\beta_t\}_{t \in \mathbb{N}_+}$.

1: Initialize $\mathcal{I}_0^{(S)} = \mathcal{I}_0^{(F)} = \emptyset$, $\theta_0 = \theta_{\max}$, $C = 0$.
2: **for** $t = 1$ to $T$ **do**
3:     $\tilde{\theta}_t \leftarrow \theta_{t-1}$.
4:     **while** $\lceil 1/\tilde{\theta}_t b(t) \rceil^d < |\mathcal{I}_{t-1}^{(F)}|$ **do**
5:         $\tilde{\theta}_t \leftarrow \tilde{\theta}_t/2$.
6:     **end while**
7:     Solve problem $\boldsymbol{x}_t = \arg\max_{\boldsymbol{x} \in \mathcal{X}} \mathrm{ucb}_t(\boldsymbol{x})$ s.t. $\forall i \in \mathcal{I}_{t-1}^{(F)}, \|\boldsymbol{x}_i - \boldsymbol{x}\|_\infty \geq \tilde{\theta}_t b(t)$.
8:     **if** solver did not return any feasible solution $\boldsymbol{x}_t$, **then**
9:         $\theta_t \leftarrow \tilde{\theta}_t/2$.
10:         $\boldsymbol{x}_t = \arg\max_{\boldsymbol{x} \in \mathcal{X}} \mathrm{ucb}_t(\boldsymbol{x})$ s.t. $\forall i \in \mathcal{I}_{t-1}^{(F)}, \|\boldsymbol{x}_i - \boldsymbol{x}\|_\infty \geq \theta_t b(t)$.
11:     **else**
12:         $\theta_t \leftarrow \tilde{\theta}_t$.
13:     **end if**
14:     **if** $c(\boldsymbol{x}_t) = 0$ **then**
15:         Observe $y_t = f(\boldsymbol{x}_t) + \epsilon_t$.
16:         Update GP by adding $(\boldsymbol{x}_t, y_t)$.
17:         $\mathcal{I}_t^{(S)} \leftarrow \mathcal{I}_{t-1}^{(S)} \cup \{t\}, \mathcal{I}_t^{(F)} \leftarrow \mathcal{I}_{t-1}^{(F)}$.
18:     **else**
19:         $\mathcal{I}_t^{(S)} \leftarrow \mathcal{I}_{t-1}^{(S)}, \mathcal{I}_t^{(F)} \leftarrow \mathcal{I}_{t-1}^{(F)} \cup \{t\}$.
20:     **end if**
21:     **if** $\sigma_{t-1}(\boldsymbol{x}_t) < h_\sigma$ **then**
22:         $C \leftarrow C + 1$.
23:         **if** $C = q$ **then**
24:             $\theta_{t-1} \leftarrow w\theta_{t-1}$.
25:             $C \leftarrow 0$.
26:         **end if**
27:     **else**
28:         $C \leftarrow 0$.
29:     **end if**
30: **end for**
31: $\hat{t} = \mathrm{argmax}_{i \in \mathcal{I}_t^{(S)}} \mathrm{lcb}_i(\boldsymbol{x}_i)$.

**Output:** $\boldsymbol{x}_{\hat{t}}$.

**Theorem F.2.** *Let $\beta_t^{1/2} = B + \sigma\sqrt{2(\gamma_{t-1;\mathcal{X}} + 1 + \ln(1/\delta))}$ and $b(t) = t^{-\alpha}$ for fixed $f \in \mathcal{H}_k$ such that $\|f\| \leq B$, $\delta \in (0,1)$, and $\alpha \in (0, d^{-1})$. Let $\boldsymbol{x}^*$ exist inside $S_c$ such that $N_{\boldsymbol{x}^*;\eta} \subset S_c$ for some $\eta > 0$. Here, we denote $N_{\boldsymbol{x}^*;\eta} = \{\boldsymbol{x} \in \mathcal{X} \mid \|\boldsymbol{x} - \boldsymbol{x}^*\|_\infty < \eta\}$. Then the following two statements hold with respect to Algorithm 4.*

    *1. $\hat{\boldsymbol{x}}_T$ exists for $T \geq \tilde{T}_s$.*

    *2. The following equation holds with probability at least $1 - \delta$:*

$$\forall T \geq \tilde{T}_s,\ r_T \leq \frac{2}{T}\left[2B(\tilde{T}_* - 1) + \sqrt{C_1 \beta_T T \gamma_{T;S_c}}\right]. \tag{94}$$

*Note that $\tilde{T}_s := \max\{(10/\overline{\theta}^d)^{1/(1-d\alpha)}, 2\} + 1$, $\tilde{T}_* := (\theta_0/\eta)^{1/\alpha} + 1$, and $\overline{\theta} := \min\{\theta_0, \eta/2, \theta_{\min}\}$ are constants that depend only on $\boldsymbol{x}^*$, $\eta$, $\alpha$, $d$, and $\theta_0$. Moreover, $C_1 = 8/\ln(1 + \sigma^{-2})$.*

**Theorem F.3.** *Let $f \in \mathcal{H}_k$ be any function such that $\|f\|_{\mathcal{H}_k} \leq B$ and fix $\delta \in (0,1)$ and $\alpha \in (0, d^{-1})$. Let $\beta_t^{1/2} = B + \sigma\sqrt{2(\gamma_{t-1;\mathcal{X}} + 1 + \ln(1/\delta))}$ and $b(t) = t^{-\alpha}$. Let $f$ be L-Lipschitz continuous (Eq. (50)) and $S_c$ satisfy Assumption 2.2. Then, the following two statements hold for Algorithm 4.*

    *1. $\hat{\boldsymbol{x}}_T$ exists for some $T \geq \tilde{T}_s$.*

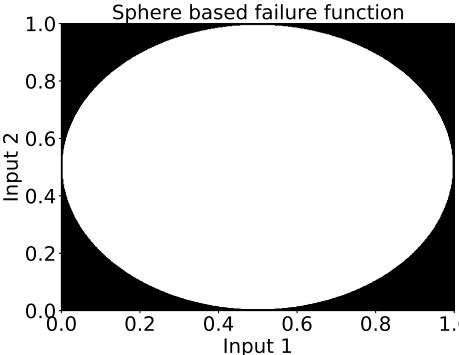
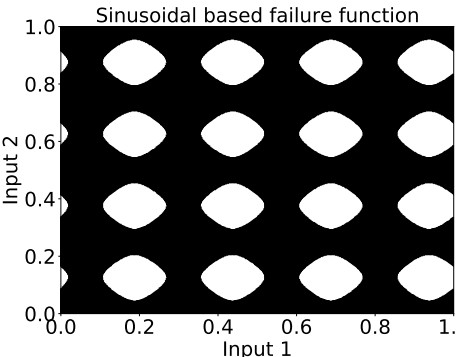

Figure 5: The failure functions used in the experiments based on the GP test functions. The shaded and unshaded regions respectively represent the success and failure regions. The function shown on the left is based on the sphere function, while the one on the right is based on sinusoidal functions.

2. *The following equation holds with probability at least $1 - \delta$:*

$$\forall T \geq \tilde{T}_s,\ r_T \leq \frac{2}{T} \left[ 2B(\tilde{T}_* - 1) + \sqrt{C_1 \beta_T T \gamma_{T;S_c}} + L\theta_0 \overline{\tau} \frac{T^{1-\alpha}}{(1-\alpha)\zeta} \right]. \qquad (95)$$

*Note that $\tilde{T}_s := \max\{(10/\overline{\theta}^d)^{1/(1-d\alpha)}, 2\} + 1$, $\tilde{T}_* := (\theta_0/\eta)^{1/\alpha} + 1$, and $\overline{\theta} := \min\{\theta_0, \zeta/2, \theta_{\min}\}$ are constants that depend only on $\boldsymbol{x}^*$, $\zeta$, $\alpha$, $d$, and $\theta_0$. Moreover, $C_1 = 8/\ln(1 + \sigma^{-2})$ and $\overline{\tau} = \|\boldsymbol{x}^* - \tilde{\boldsymbol{x}}\|_\infty$.*

Theorem F.2 immediately follows from Theorem A.6 by noting that the following holds under Algorithm 4.

1. $\theta_t b(t)$ is monotone decreasing.
2. $\forall t \in \mathbb{N}_+, \theta_t \geq \min\{\theta_0, \eta/2, \theta_{\min}\}$.

First, Lemma A.7 holds under the monotonicity of $\theta_t b(t)$. From statement 2 above, Lemma A.8 holds by substituting the lower bound of $\theta_{T_*}$ with $\min\{\theta_0, \eta/2, \theta_{\min}\}$. Lemma A.11 also holds under this condition. Consequently, the same argument holds under Algorithm 4 where $\overline{\theta}$ is replaced by $\min\{\theta_0, \eta/2, \theta_{\min}\}$ in Theorem A.6. Theorem F.3 also holds likewise.

# G  Additional experiments

## G.1  2D-GP-generated Test Functions

We perform experiments using synthetic objective functions generated from GP. First, we generate 100 random points over $\mathcal{X} := [0, 1]^2$. Then, we sample function values on those 100 points from the GP prior distribution. The objective function to be tested is the posterior mean computed by feeding the pairs of the generated input points and the function values to the same GP. Following the above process, we generate 5 test functions. For each test function, experiments are performed 10 times, with a total of 50 experiments. The average performance is evaluated. We use the GP with a prior mean of 0 and the Gaussian kernel $k(\boldsymbol{x}, \boldsymbol{y}) := \sigma_f^2 \exp(-\|\boldsymbol{x} - \boldsymbol{y}\|^2/(2l_f^2))$ where $\sigma_f = 1$ and $l_f = 0.2$ for the generation of $f$ and the modeling within the algorithm. We use two types of failure functions $c$ with the success regions depicted in Fig. 5. We refer to them respectively as the sphere and sinusoidal failure functions. The mathematical definitions are given in Appendix H.

Figure 6 shows the result of the experiments. We note that, in the EI and GP-UCB algorithms, the regret stops decreasing in the early stage. This occurs by getting stuck on the same failure point because failures are not considered by those algorithms. In the case of the sphere failure function, F-GP-UCB performs slightly worse compared to the other GPC-based algorithms. It is reasonable to expect that model-based algorithms perform well on simple failure functions such as

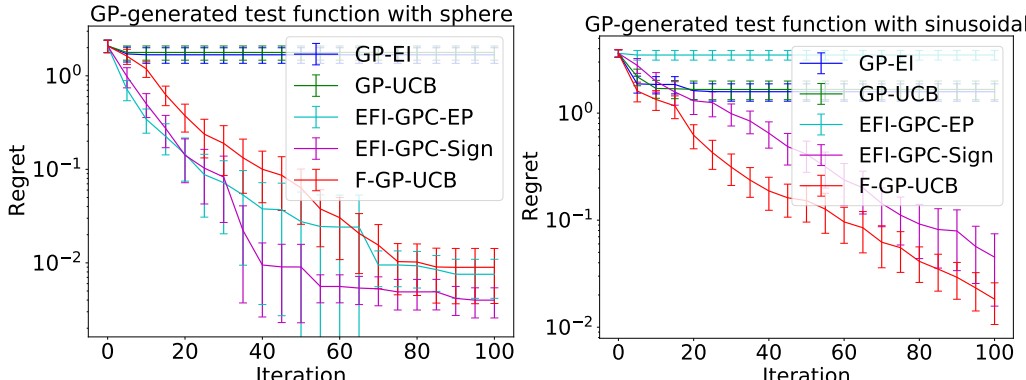

Figure 6: The evolution of the regret in the synthetic experiments based on the GP test functions. The left plot uses the sphere failure function. The right plot uses the sinusoidal failure function. The results show the average of 20 experiments, each with a different random seed. The error bars correspond to two standard errors.

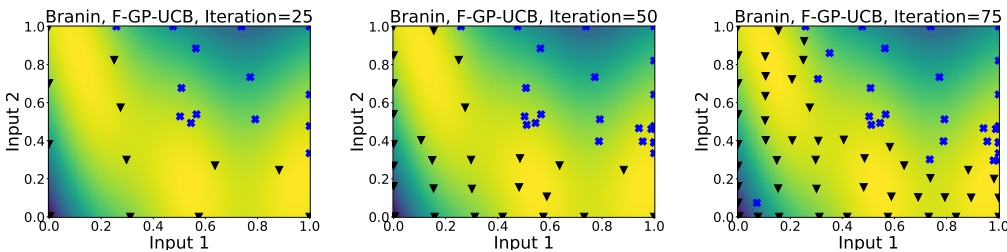

Figure 7: The behavior of F-GP-UCB is shown at iteration numbers 25 (left), 50 (middle), and 75 (right), respectively, for a fixed random seed. The observed points are overlaid over the objective function: failures are shown as black triangles while successes are shown as blue crosses.

the sphere function which can be modeled with ease. The final regret of F-GP-UCB is comparable to that of EFI-GP-EP. Thus no crucial differences in the performance are seen. For the sinusoidal failure function, F-GP-UCB has a better performance than the other algorithms. In particular, in the EFI-GP-EP algorithm, the probabilistic GPC has a serious misspecification in the model parameter. Namely, the length scale parameter $l_c$ is excessively large, while the output scale parameter $\sigma_c$ is excessively small.[6] It is observed that these parameter misspecifications lead to the next points getting stuck on the same failure points in the early stage. Note that the risk of misspecifying kernel parameters in classic GPC is a drawback of the EFI-GP-EP algorithm as also mentioned in [2].

### G.2 Behavior of the F-GP-UCB algorithm in Branin problem

In the experiment of our test problem using the Branin function, the ideal search strategy is to control the trade-off between exploration and exploitation in the upper right region which reduces the regret, while also continuing to seek the isolated success regions in the rest of the search space where there are mostly failures.

Figure 7 shows the snapshots of the observed points at different iterations for a fixed random seed. From Fig. 7, it can be seen that F-GP-UCB makes balanced observations between the large success region on the upper right side and the rest of search space which consists of mostly failures.

---

[6]The exact parameter values are given in Appendix H.

# H   Details of numerical experiments

## H.1   Details of benchmark functions.

We give the details of the benchmark functions used in the numerical experiments in Sec. 6 and Appendix G.

**Sphere and sinusoidal failure function.**   We describe the sphere and sinusoidal failure functions used in the experiments based on GP test functions. The sphere failure function $c_{\text{sphere}} : [0,1]^2 \to \{0,1\}$ is defined as follows.

$$c_{\text{sphere}}(\boldsymbol{x}) = \mathbb{1}\{\|2\boldsymbol{x} - 1\|_2^2 > 1\}. \tag{96}$$

Here, $\mathbb{1}\{\cdot\}$ is an indicator function that gives 1 when the argument is a true statement and returns 0 otherwise. The sinusoidal failure function $c_{\text{sinusoidal}} : [0,1]^2 \to \{0,1\}$ is defined as follows.

$$c_{\text{sinusoidal}}(x_1, x_2) = \mathbb{1}\{g(2\boldsymbol{x} - 1) > -1.5\}, \tag{97}$$

where $g$ is a constraint function used in [13] and is given as

$$g(x_1, x_2) = \sin(4\pi x_1) - 2\sin^2(2\pi x_2). \tag{98}$$

**Branin test problem.**   The Branin function, with its input space scaled to $[0,1]^2$, is used as the objective function in the test problem. We define the objective function $f : [0,1]^2 \to \mathbb{R}$ as

$$f(x_1, x_2) = -f_{\text{Branin}}(15x_1 - 5, 15x_2), \tag{99}$$

where $f_{\text{Branin}} : [-5, 10] \times [0, 15] \to \mathbb{R}$ is given as follows:

$$f_{\text{Branin}}(x_1, x_2) = \left(x_2 - 5.1x_1^2 + \frac{5}{\pi}x_1 - 6\right)^2 + 10\left(1 - \frac{1}{8\pi}\right)\cos(x_1) + 10. \tag{100}$$

The failure function $c : [0,1]^2 \to \{0,1\}$ is given as

$$c(\boldsymbol{x}) = \mathbb{1}\{g(2\boldsymbol{x} - 1) > 0\}, \tag{101}$$

where $g(\boldsymbol{x})$ is given as follows:

$$g(x_1, x_2) \tag{102}$$
$$= \min\{(x_1 - 1)^2 + (x_2 - 1) - 2 - 1.5^2, \tag{103}$$
$$(x_1 - \frac{2(\pi + 5)}{15} - 1)^2 + (x_2 - \frac{4.55}{15} - 1)^2 - 0.1^2, \tag{104}$$
$$(x_1 + 0.9)^2 + (x_2 + 0.9)^2 - 0.1^2, \tag{105}$$
$$(x_1 + 0.6)^2 + (x_2 + 0.6)^2 - 0.1^2\}. \tag{106}$$

**Gardner test problem.**   The test problem is based on the benchmark function used in [10]. The objective function $f : [0,1]^2 \to \mathbb{R}$ and the failure function $c : [0,1]^2 \to \{0,1\}$ are respectively defined as

$$f(\boldsymbol{x}) = -f_{\text{Gardner}}(6\boldsymbol{x}), \ c(\boldsymbol{x}) = \mathbb{1}\{g_{\text{Gardner}}(6\boldsymbol{x}) > 0.5\}, \tag{107}$$

where $f_{\text{Gardner}}$ and $g_{\text{Gardner}}$ are given as follows:

$$f_{\text{Gardner}}(x_1, x_2) = \cos(2x_1)\cos(x_2) + \sin(x_1), \tag{108}$$
$$g_{\text{Gardner}}(x_1, x_2) = \cos(x_1)\cos(x_2) - \sin(x_1)\sin(x_2) + 0.5. \tag{109}$$

**Hartmann test problem.** The test problem uses the benchmark function in [29] which is modified for three dimensions. The objective function $f : [0,1]^3 \to \mathbb{R}$ is given as follows:

$$f(\mathbf{x}) = \sum_{i=1}^{4} \alpha_i \exp\left(-\sum_{j=1}^{3} A_{ij}\left(x_j - P_{ij}\right)^2\right), \quad \text{where} \tag{110}$$

$$\alpha = (1.0, 1.2, 3.0, 3.2)^\top, \tag{111}$$

$$\boldsymbol{A} = \begin{pmatrix} 3.0 & 10 & 30 \\ 0.1 & 10 & 35 \\ 3.0 & 10 & 30 \\ 0.1 & 10 & 35 \end{pmatrix}, \tag{112}$$

$$\boldsymbol{P} = 10^{-4} \begin{pmatrix} 3689 & 1170 & 2673 \\ 4699 & 4387 & 7470 \\ 1091 & 8732 & 5547 \\ 381 & 5743 & 8828 \end{pmatrix}. \tag{113}$$

The failure function $c : [0,1]^3 \to \{0,1\}$ is defined as follows:

$$c(\boldsymbol{x}) = \mathbb{1}\{\|\boldsymbol{x}\|_2^2 > 1\}. \tag{114}$$

## H.2 Details of numerical experiments of quasicrystals.

In these numerical experiments, we define the objective function $f : [0,1]^2 \to \mathbb{R}$ to be maximized as follows:

$$f(x_1, x_2) = -\left(\frac{1}{0.84623\hat{x}_1 + 0.646994\hat{x}_2 + 1.20782}\right) - 0.3328 \tag{115}$$

where $\hat{x}_1 = 0.15x_1 + 0.15$ and $\hat{x}_2 = 0.15x_2 + 0.05$. We describe the failure function after we give the motivation for the choice of the objective function.

The above formula is based on the consideration of the quasicrystal properties in the Al–Cu–Mn ternary system. We consider the relative composition of the three elements $x_{\mathrm{Al}}, x_{\mathrm{Cu}}, x_{\mathrm{Mn}} \in [0,1]$ which are required to sum to unity: $x_{\mathrm{Al}} + x_{\mathrm{Cu}} + x_{\mathrm{Mn}} = 1$. Because of this constraint, the number of degrees of freedom is two. Then, we define a heuristic function $k_{\mathrm{ph}} : \mathbb{R} \to \mathbb{R}$ that models the phonon thermal conductivity of quasicrystals based on real data [46].[7] We take Fig. 6 of their paper and use the fitted curve as the heuristic function $k_{\mathrm{ph}}(A_{\mathrm{mean}})$ which takes as input the mean atomic weight of the alloy $A_{\mathrm{mean}}$. Our extraction of the heuristic function from the paper is

$$k_{\mathrm{ph}}(A_{\mathrm{mean}}) = \left(\frac{1}{0.02314 A_{\mathrm{mean}} + 0.5835}\right) + 0.3328. \tag{116}$$

In our Al-Cu-Mn ternary system, $A_{\mathrm{mean}}$ is calculated as follows:

$$A_{\mathrm{mean}} = x_{\mathrm{Al}} A_{\mathrm{Al}} + x_{\mathrm{Cu}} A_{\mathrm{Cu}} + x_{\mathrm{Mn}} A_{\mathrm{Mn}}. \tag{117}$$

where the individual atomic weights are $A_{\mathrm{Al}} = 26.98$, $A_{\mathrm{Cu}} = 63.55$, and $A_{\mathrm{Mn}} = 54.94$. The objective of this experimental setup is to minimize $k_{\mathrm{ph}}$ for quasicrystals. Thus we identify the objective function to be $f = -k_{\mathrm{ph}}$. We pick $\hat{x}_1 := x_{\mathrm{Cu}}$ and $\hat{x}_2 := x_{\mathrm{Mn}}$ as the independent fractions. This allows us to substitute $x_{\mathrm{Al}} := 1 - \hat{x}_1 - \hat{x}_2$. Furthermore, we restrict the search space to $\hat{x}_1 \in (0.15, 0.3)$ and $\hat{x}_2 \in (0.05, 0.2)$, because, without this, the success region is too small, resulting in too small differences in the comparison of the methods. This corresponds to the experimental setup that that the phase boundaries of quasicrystal formation are roughly known, but not precisely. Finally, we define $x_1 := (\hat{x}_1 - 0.15)/0.15$ and $x_2 := (\hat{x}_2 - 0.05)/0.15$ for input normalization so that the search space becomes $(x_1, x_2) \in [0,1]^2$. Substituting these definitions into Eqs. (116)–(117) yields Eq. (115).

The success region is defined to be the region where the quasicrystal forms in the Al-Cu-Mn ternary system using data [16], from which we take the $D_3$ phase in their main result to be the success region. Outside of this region where quasicrystals do not form and is defined to be the failure region. Figure 8 shows the success and failure regions together with the distribution of $k_{\mathrm{ph}}$.

---

[7] In ordinary crystals, the thermal conductivity has a negative temperature coefficient, i.e., the conductivity decreases at higher temperatures. Quasicrystals have unusually low thermal conductivity with a positive temperature coefficient. In certain applications, this property of positive temperature coefficient is desired, and departure from this property is therefore considered a failure.

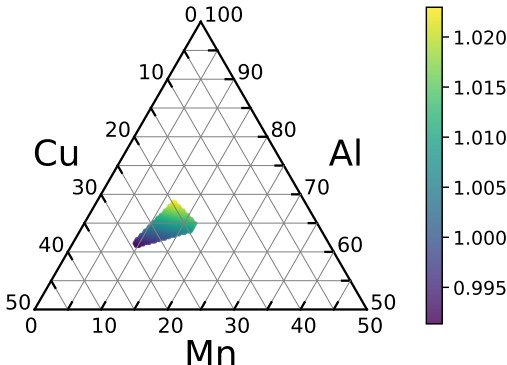

Figure 8: Triangular plot with the axes corresponding to the relative composition of the elements in the alloy, given in percent. The colored region corresponds to the success region, while the uncolored region corresponds to the failure region. The color hue is given for the phonon thermal conductivity of quasicrystals in units of $\mathrm{W\,m^{-1}\,K^{-1}}$. We adapted the plotting script from [33].

## H.3 Details of other algorithms and implementation.

In this section, we describe the implementation of the numerical experiments and comparison methods. In all the methods, we used BoTorch [3] for the GP regression of the objective function $f$ and the optimization of kernel hyperparameters. In all the numerical experiments (except the GP test function setup), the Gaussian kernel hyperparameters $\sigma_f$ and $l_f$ of the GP model are fixed beforehand using the following procedure. First, we generate a Sobol sequence of 1024 points over $\mathcal{X}$. Then, the objective function is evaluated over the generated Sobol sequence. The obtained pairs of input and output are used to train the GP model. We select $\sigma_f$ and $l_f$ values that maximize the marginal likelihood which are used for all the methods.

Next, we describe the details of comparison methods.

**GP-EI.** Using a GP model constructed from only the successful observations, the point with the maximum EI is selected. We define the observed point $\boldsymbol{x}_t$ at each step as $\boldsymbol{x}_t = \mathrm{argmax}_{\boldsymbol{x}\in\mathcal{X}}\mathrm{EI}_t(\boldsymbol{x})$, $\mathrm{EI}_t(\boldsymbol{x}) = \mathbb{E}_t[\max\{f(\boldsymbol{x}) - \tilde{y}_t, 0\}]$. Here, the expected value is taken with respect to the GP posterior distribution. Let $\{y_i\}_{i\in\mathcal{I}_t^{(S)}}$ be the observed values up to step $t$. We define $\tilde{y}_t$ to be the maximum value among the observed values. Furthermore, if $\mathcal{I}_t^{(S)} = \emptyset$, we define $\tilde{y}_t = 0$. We note that, since we fix the prior mean to 0 regardless of the input, there are essentially no differences in the choice of the imputation value in the case of no observations. Finally, the standard BoTorch acquisition function solver was used for choosing $\boldsymbol{x}_t$.

**GP-UCB.** We observe the point over $\mathcal{X}$ that maximizes $\mathrm{ucb}_t(\boldsymbol{x})$. That is, we let $\boldsymbol{x}_t = \mathrm{argmax}_{\mathrm{x}\in\mathcal{X}}\{\mu_{t-1}(\boldsymbol{x}) + \beta_t^{1/2}\sigma_{t-1}(\boldsymbol{x})\}$, where $\beta_t = 2\ln(2t)$ is used in the experiments. As in GP-EI, the BoTorch solver function is used to optimize the selection of the input point.

**EFI-GPC-EP.** This method uses a GPC model with a probit likelihood to model the failure function. We use the EP method [36] to approximate the computation of the GPC posterior distribution. In this method, the observed point at each step is given as $\boldsymbol{x}_t = \mathrm{argmax}_{\boldsymbol{x}\in\mathcal{X}}p_t(\boldsymbol{x})\mathrm{EI}_t(\boldsymbol{x})$, where $p_t(\boldsymbol{x})$ is the prior success probability from GPC. The GPC kernel hyperparameters $\sigma_c$ and $l_c$ are fixed in the same way as the kernel hyperparameters for $f$, i.e., based on marginal likelihood optimization using a Sobol sequence of 512 points over $\mathcal{X}$. The open-source software GPy [15] was used for the GPC EP inference and the marginal likelihood optimization. To select the $\boldsymbol{x}_t$ optimization, the Dividing Rectangles (DiRect) algorithm [23] implemented using the NLopt package [22] is used. The maximum number of evaluations by DiRect is set to 500.

**EFI-GPC-Sign.** This is a deterministic GPC method proposed by [2]. We use the latent function $g \sim \mathcal{GP}(0, k_c)$ with $c(\boldsymbol{x}) = \mathbb{1}\{g(\boldsymbol{x}) > 0\}$ to model the deterministic GPC, where $\mathcal{GP}(\mu, k)$ is a GP with a mean function $\mu$ and covariance function $k_c$. The original paper treats the prior mean

Table 1: The Gaussian kernel hyperparameters used in the numerical experiments. The values are chosen based on the marginal likelihood. The columns correspond to the objective function $f$, EFI-GPC-EP's classical GPC, and EFI-GPC-Sign's deterministic GPC model. For the GP test function experiment, the objective functions' kernel hyperparameters are not listed because they are fixed.

|  | Objective function | EFI-GPC-EP | EFI-GPC-Sign |
|---|---|---|---|
| Sphere |  | $\sigma_c^2 = 13.4, l_c = 0.29$ | $\sigma_c^2 = 1.0, l_c = 0.2$ |
| Sinusoidal |  | $\sigma_c^2 = 0.26, l_c = 4.13$ | $\sigma_c^2 = 1.0, l_c = 0.2$ |
| Branin | $\sigma_f^2 = 110148, l_f = 0.30$ | $\sigma_c^2 = 5.05, l_c = 0.199$ | $\sigma_c^2 = 1.0, l_c = 1.0$ |
| Gardner | $\sigma_f^2 = 8.47, l_f = 0.26$ | $\sigma_c^2 = 7.63, l_c = 0.199$ | $\sigma_c^2 = 1.0, l_c = 0.2$ |
| Hartmann | $\sigma_f^2 = 0.46, l_f = 0.20$ | $\sigma_c^2 = 5.87, l_c = 0.410$ | $\sigma_c^2 = 1.0, l_c = 0.8$ |
| Quasicrystal | $\sigma_f^2 = 0.03, l_f = 4.85$ | $\sigma_c^2 = 5.52, l_c = 0.19$ | $\sigma_c^2 = 1.0, l_c = 0.2$ |

function as an adjustable parameter. In this paper, we fix the prior mean to $0$ for simplicity. As is the case for ordinary GPC, this model also requires an approximation method as the posterior distribution of $g$ cannot be computed analytically. In this paper, we employ Gibbs sampling where the burn-in and thinning periods are set to be $1000$ and $10$, respectively. Then the posterior success probability is approximated based on the $100$ approximate samples of posterior distributions of $g$. The approximated posterior success probability $p_t(\boldsymbol{x})$ is used to choose $\boldsymbol{x}_t$. Specifically, as in EFI-GPC-EP, we choose $\boldsymbol{x}_t = \mathrm{argmax}_{\boldsymbol{x} \in \mathcal{X}} p_t(\boldsymbol{x}) \mathrm{EI}_t(\boldsymbol{x})$. For the calculation of the marginal likelihood, we need the multivariate normal orthant probability with the same number of dimensions as the training sample size. In this paper, we use the approximation method for the orthant probability based on [12]. The kernel hyperparameter is chosen by maximizing the marginal likelihood over a Sobol sequence of $32$ points over $\mathcal{X}$. In particular, as the sample size over $\mathcal{X}$ becomes large, the orthant probability becomes excessively small and causes numerical precision problems. Therefore, for the marginal likelihood calculation, we choose a relatively small Sobol sequence with $32$ points. For simplicity, we fix $\sigma_c = 1$ and choose $l_c$ by doing a grid search in the range $[0.2, 2.0]$. The grid size is set to $10$. Finally, $\boldsymbol{x}_t$ is chosen using DiRect, as is EFI-GPC-EP.

**F-GP-UCB.** The basic parameter settings are fixed and described in Sec. 5. That is, we let $b(t) = t^{-1/2d}, w = 0.75, h_\sigma = 0.02, q = 3, \theta_{\min} = 0.0001, \theta_{\max} = 0.5, \beta_t = 2\ln(2t)$. To choose $\boldsymbol{x}_t$, we employ the following process. First, a Sobol sequence with $1024$ points is generated as candidates for the initial points. Among the $1024$ points, if a feasible point is found inside $\mathcal{X}_t$, we select up to $5$ points in the order of highest $\mathrm{ucb}_t(\boldsymbol{x})$ value. These points are used as initial points. Then, we run the optimizations by using the method of moving asymptotes [45] implemented in NLopt.

If there are fewer than $5$ feasible points among the $1024$ points, we select the non-feasible points in the order of highest $\mathrm{ucb}_t(\boldsymbol{x})$ value as initial points, which are used as initial points for the augmented Lagrangian method in NLopt. We set the maximum number of evaluations in the augmented Lagrangian method to $100000$. The limited-memory BFGS method is used as the inner solver. If no feasible points are obtained from this process, the scale parameter $\theta_{t-1}$ is multiplied by $1/2$, and the optimization is repeated.

The parameter values $\sigma_f, l_f, \sigma_c$, and $l_c$ fixed by marginal likelihood optimization in each experiment are summarized in Tab. 1.