# OpenReview forum: "Failure-Aware Gaussian Process Optimization with Regret Bounds"
_NeurIPS.cc/2023/Conference — NeurIPS 2023 poster_

### Official Review · Reviewer_ven9 · 2023-07-05

**Soundness:** 3 good
**Presentation:** 3 good
**Contribution:** 3 good
**Rating:** 6
**Confidence:** 4

**Summary:**

1. A failure-aware gaussian process optimization algorithm is proposed. A key difference from existing GP constrained optimization is that here we cannot obtain a direct observation from the latent failure function.
2. The algorithm's regret bound is presented.
3. Some practical issues are discussed.
4. Numerical experiments are conducted, the proposed algorithm shows superiority on some benchmarks.


**Strengths:**

1. Problem formulation is novel.
2. Theoretical derivation is solid
3. Practical issues are considered and numerical experiments show its potential.

**Weaknesses:**

1. Learning feasible set without knowing failure function is a key feature of paper's problem formulation. The NP-complete set covering problem is assumed to be fully solved in theoretical analysis while numerical experiments adopt a heuristic approximation algorithm.  It seems there is a gap between theoretical part and numerical part.
2. Algorithm doesn't show dominant superiority on all all numerical benchmarks.


**Questions:**

Can we have a more "rigorous" theoretical analysis of the "real" algorithm(replacing set covering part by its heuristic approximation) and still derive the regret bound?

**Limitations:**

No potential negative social impact is seem.

---

> ### Author Rebuttal · Authors · 2023-08-09
>
> We thank the reviewer for the comments.
>
> > Can we have a more "rigorous" theoretical analysis of the "real" algorithm(replacing set covering part by its heuristic approximation) and still derive the regret bound?
>
> We consider the input domain of bounded hypercube, on which the NP-complete set-covering problem is needed to solve, as one of the most common settings of the GP-optimization field. Another common setting is a discrete input domain $\mathcal{X}$, on which set-covering problems are easily solved by brute force way with $\mathcal{O}(d|\mathcal{X}|^2)$ computational costs.
> We conjecture that by combining discretization arguments used in previous works (e.g., [1, 2, 3]), our algorithm can be extended to a general continuous input domain without breaking the theoretical guarantee so that the set-covering problem on the original continuous input domain is not required to solve.
> This direction of research is interesting, and we leave it as future work.
>
>
> [1]Chowdhury et al., "On kernelized multi-armed bandits.", 2017
>
> [2]Vakili et al., "Optimal order simple regret for Gaussian process bandits.", 2021
>
> [3]Li et al., "Gaussian process bandit optimization with few batches.", 2022

---

> > ### Comment · Reviewer_ven9 · 2023-08-18
> >
> > Thanks for the reply. I will keep my original assessment.

---

### Official Review · Reviewer_Yqeo · 2023-07-06

**Soundness:** 4 excellent
**Presentation:** 2 fair
**Contribution:** 3 good
**Rating:** 7
**Confidence:** 4

**Summary:**

This paper studies Bayesian optimization in the setting where observing the unknown function might fail, for instance due to numerical simulator returning NaN. The authors propose a UCB-type acquisition function for this setting, which is based on first restricting the domain to a feasible regions where failures are less likely to occur, and then applying standard UCB-based acquisition function constrained to that region. The authors present a bound on the simple regret in this setting, and perform some experiments which consider only low-dimensional examples but show good performance compared to baselines.

EDIT: I've updated my score in light of the rebuttal.

**Strengths:**

The paper's topic is important and of key relevance for practical Bayesian optimization, where observation failure is a common issue which is mostly addressed using ad-hoc methods. They have clearly described their setting and distinguished it from others, and everything is properly cast into a well-defined mathematical problem, which is clearly described.

The introduction is well-written and does a good job of introducing the reader to both the setting and the authors' approach, and gives due credit to prior work.

Figure 1 is conceptually very nice and provides a good illustration that makes it easy for the reader to see early on what is going on: the authors' algorithm splits the state space into a region where observations are likely to work, and another region where they are unlikely to work. One request: **please make the fonts bigger to match the font size of the text**, so readers with not the best eyesight can actually see everything.

The authors do a good job at focusing on their part of the contribution, and don't take up lots of space stating obvious corollaries of their result, but at the same time also do a good job at stating what else can be done, for instance by combining their analysis with prior work such as RBF specific bounds in the setting without failures.

Overall, I think this is a good paper with various fixable flaws described below, so I don't have too many additional comments.

**Weaknesses:**

Details of the proposed algorithm are unclear. From my reading of the paper, I understand that what you do is define a "failure-safe space" consisting of the full space minus a set of infinity-balls around points where failure occurred. Then, at certain points in time whose definition I still don't fully grasp, you cut the radius of the balls by half, so that points which were previously excluded from consideration because they were too close to a failure are now considered again. Is this correct? If so, this needs to be stated much more explicitly and much more clearly, using an intuitive description in addition to equations or notation-heavy pseudocode. If not, please correct my understanding.
* In particular, please provide a more intuitive explanation of what \theta_t is doing, and in what cases it gets cut by half - I am still confused how when this happens

Experiments
* Unless I have missed something, the highest dimension problem used in experiments is three. This is very limiting compared to much of the literature and much smaller than many problems of practical interest
* The quasicrystal experiment is very light on details, in spite of being a really cool illustration. This section deserves significantly more text on both the setup and results so that readers can understand and contextualize what is going on. The fact that this problem is, unless I am mistaken, three-dimensional, could also have been stated more explicitly

In the analysis, the regret bound presented seems fairly specific, since it is talking only about simple rather than cumulative regret, and has a fairly specific way through which the point returned is selected. In bandits at least this kind of setting is certainly out there, but not something I see in every paper, so I'm a bit unsure as to whether the ideas are general or very specific to this setting.
* As a result, I'm curious: how strongly does the theoretical analysis and empirical performance depend on the specific way through which the point returned is selected?

Writing: the paper tends to go back and from between parts that were very clear and well-written, and parts where I was confused. Part of this is due to typos and grammar, which is fixable. However, there are other times where grammar is not the issue, and there are many points where the paper would be much better if it provided helpful hints to the reader to orient their reading in the right way to avoid them becoming confused. Detailed presentation comments below:
* The paper has slightly too many acronyms, for instance GPC, which is not an important enough concept in this paper to make it into an acronym and should instead be spelled out
* There's slightly too much discussion on prior work, which takes up space that is needed elsewhere - I'd prefer to see this part tightened up
* When I first read this, I was confused about definition of regret, namely why is r_t = f(x^*) - min_{x\in X} f(x) not equal to zero, but then I realized this is because the authors are interested in reward maximization (bandit style) rather than cost minimization (online learning style). A line which saves the reader time by clarifying this should be added.
* Aesthetic: curly brace notation for the posterior is really ugly - consider using parentheses
* **Please number all equations, not just the ones you think are important.** The convention of only numbering important equations is archaic, outdated, and harmful to readers. Numbering everything prevents readers from having to ask about "the second unnumbered equation on page 2", and then the person who was asked is confused whether it's the second equation from the top or from the bottom. Doing so will significantly improve the paper.
* Why should the logdet be called the maximum information gain? If you're going to use these terms, you should connect this with mutual information or some kind of related quantity. This choice of terminology should be explained rather than just parachuted in.
* Since the notion of "interior point" is introduced in the definition directly below it, maybe the sentence should include "introduced below" or similar.
* Assumption 2.2: please add a textual note that this is an infinity-ball, not a Euclidean ball, so that someone who misses the infinity symbol doesn't get confused
* Proposed algorithm could be replaced with a more informative section title
* No need to introduce packing numbers with too much detail, the audience of this paper will know about that

**Questions:**

Line 94: what do you mean by "uses the modeling information of f"? I didn't understand this sentence

I'm confused about whether or not you assume you know the reproducing kernel of the RKHS the unknown function lies in explicitly, or not. Does the kernel of the unknown function need to match the kernel used for the GP prior? I think you assume it's known, which is reasonable - please let me know if that's correct and please make this more explicit in the text.

Theorem 4.2: should there be a "with probability 1-\delta" quantifier for "the estimated solution exists .."? What stops the algorithm from getting arbitrarily unlucky and obtaining no data at arbitrarily large times? Does this have to do with the assumption that simulator failure is deterministic?

Typos and English grammar fixes below:
* Line 40: "converges the optimal" -> "converges to the optimal"
* Line 47: what does it mean "has a close relation to this study"?
* Line 52: "under the settings" -> "under the setting"
* Line 54: "that a noisy observation" -> "that noisy observation"
* Line 58: "modifying the settings of our study" -> unsure what this sentence means
* Line 60: "that takes into account failures" -> "that take into account failures"
* Line 63: "that the observation failure" -> "that observation failure"
* Line 65: "as described in [2]" -> "as described in Bachor et al. [2]"
* Line 70: "one latent function. (e.g. ..." -> "one latent function (e.g. ..."
* Line 78: "which is a conditional" -> "which is conditionally"
* Line 117: "each of its observation" -> "each of its observations"
* Line 122: "The regret upper bound cannot be obtained" -> "A regret upper bound cannot be obtained"
* Line 247: "Practical consideration" -> "Practical considerations"

**Limitations:**

*Almost* adequately - the main thing that is missing is a more comprehensive discussion on the dependence of their proposed methods on the dimension of the unknown function's domain.

---

> ### Author Rebuttal · Authors · 2023-08-09
>
> We thank the reviewer for the detailed comments.
> In the revised version of our paper, we will carefully fix the typos, grammar mistakes, and clarity issues, which the reviewer kindly pointed out.
> We describe the answers to the questions of the reviewer below.
>
>
> > Then, at certain points in time whose definition I still don't fully grasp, you cut the radius of the balls by half, so that points which were previously excluded from consideration because they were too close to a failure are now considered again.
> Is this correct?
>
> We cut the radius of the balls by half when the algorithm detects that previous failure points exclude overall search space.
> (That is, $\mathcal{X}_t$ defined on Eq.(4) becomes an empty set.)
> The monotone decreasing function $b$ has a role in controlling so that points that were previously excluded from consideration because they were too close to a failure are considered again.
> The detail is described in Lines 178-185.
>
>
> > In particular, please provide a more intuitive explanation of what $\theta_t$ is doing, and in what cases it gets cut by half - I am still confused how when this happens
>
> The parameter $\theta_t$ has a role in avoiding the event that $\mathcal{X}_t$ becomes an empty set.
> If we do not halve $\theta_t$ at all times (that is $\theta_t = \theta_0$ at all times), this event might occur, especially when the user sets $\theta_0$ without prior knowledge about the failure function in practice.
> For example, consider 1 dimensional problem with $\mathcal{X} = [0, 1]$, $\theta_0 = 0.2$, and $b(t) = (1/t)^{1/2}$.
> Moreover, let us assume that success region $S_c = [0.0001, 0.0002]$, and the user chooses $x_1$ as $x_1 = 0.0$.
> In this extreme case, $\mathcal{X}_t$ becomes empty set at least $t=25$ since $25$ is 0.04-packing number of $[0, 1]$ (the radius at $t=25$ is $0.04:= (0.2/\sqrt{25})$, and all $x_t$ will fail before $t=25$).
> Intuitively, the event that $\mathcal{X}_t$ becomes an empty set will occur if the user sets $\theta_0$ with too large value (or sets $b(t)$ as too slow decreasing rate function) in contrast to the size of success region.
> By halving \theta_t when $\mathcal{X}_t$ is detected to be an empty set, our algorithm could define non-empty search space $\mathcal{X_t}$ and choose $x_t$ at any time step without knowing unknown failure regions.
>
>
> > In the analysis, the regret bound presented seems fairly specific, since it is talking only about simple rather than cumulative regret, and has a fairly specific way through which the point returned is selected.
> In bandits at least this kind of setting is certainly out there, but not something I see in every paper, so I'm a bit unsure as to whether the ideas are general or very specific to this setting.
>
> As far as we know, the estimated solution based on a lower confidence bound, such as Eq.(6), is commonly used in the GP-optimization field (e.g., [1, 2]).
> The only difference between our setup and standard settings is that the index $\hat{t}$ of Eq.(6) is defined over the previous success index set $I_t^{(S)}$.
> (The index $\hat{t}$ of an estimated solution is defined over the overall time index set ${1, \ldots, t}$ in the standard setting.)
> We believe that our definition of $\hat{x}_t$ is a natural extension of previous works and not too specific to our setting.
> (As described in a footnote of page 3, it should be noted that the cumulative regret upper bound of our algorithm is also provided in Appendix C.)
>
> [1]Bogunovic et al., Adversarially robust optimization with Gaussian processes, 2018
>
> [2]Kirschner et al., Distributionally robust Bayesian optimization, 2019
>
>
> > how strongly does the theoretical analysis and empirical performance depend on the specific way through which the point returned is selected?
>
> As described in the earlier answer, the only specific point of our estimated solution is that the index $\hat{t}$ of Eq.(6) is defined over the previous success index set $I_t^{(S)}$ without using failed index set. If we use failed index set to define $\hat{t}$, the estimated solution might be a failure point (that is, worst-case regret will occur).
>
>
> > Line 94: what do you mean by "uses the modeling information of f"? I didn't understand this sentence
>
> This sentence simply means that our algorithm uses the posterior mean and variance of the GP modeling of $f$ (e.g., Eq.(5), Eq.(6)).
>
>
>
> > I'm confused about whether or not you assume you know the reproducing kernel of the RKHS the unknown function lies in explicitly, or not.
> Does the kernel of the unknown function need to match the kernel used for the GP prior? I think you assume it's known, which is reasonable - please let me know if that's correct and please make this more explicit in the text.
>
> Yes, we assume that a kernel $k$, which corresponds to RKHS $\mathcal{H}_k$ the function $f$ lies, is known.
> Furthermore, we assume that the kernel used in GP matches the kernel of $\mathcal{H}_k$. We will make these assumptions more explicit in Sec.2 in the revision.
>
>
> > Theorem 4.2: should there be a "with probability 1-\delta" quantifier for "the estimated solution exists .."?
> What stops the algorithm from getting arbitrarily unlucky and obtaining no data at arbitrarily large times?
>
> The guarantee for the existence of the estimated solution in Theorem 4.2 does not require probabilistic arguments.
> Under Assumption 2.2, our algorithm design stops the algorithm from getting arbitrarily unlucky and obtaining no data at arbitrarily large times.
> Intuitively, as described in Lines 142-152, our algorithm is carefully designed to cover overall input space without breaking the regret guarantee.
> Therefore, if there exists a success region whose volume is strictly greater than zero (which is assumed in Assumption 2.2), the algorithm eventually observes the success point within finite time steps.

---

> > ### Comment · Reviewer_Yqeo · 2023-08-14
> >
> > Thanks for these explanations, that makes sense. My general sense is that it would be good to have more papers on this topic to stimulate further work. I bumped my score, but, please number all of your equations, since in a paper like this numbers are extremely useful for the readers.

---

### Official Review · Reviewer_Mr4k · 2023-07-11

**Soundness:** 3 good
**Presentation:** 2 fair
**Contribution:** 3 good
**Rating:** 5
**Confidence:** 3

**Summary:**

The paper extends the gaussian process UCB algorithm to the case where some observations are failures. i.e. They solve for the case where x^* = argmax_{x in X} {f(x) such that c(x) =0}. Here c(x) = 1 may be thought of as an indicator of failure. The crucial part is to adaptively ignore the neighbourhood of the failure point to choose the next evaluation point. The paper does a number of experiments to show that their algorithm works with respect to the claims made.

**Strengths:**

- Excellent numerical experiment on quasicrystals. This was quite novel to consider the Al-Cu-Mn crystal system.
- The idea of applications of GP-UCB to account for failures of observation was quite novel.
- The notation of failure as a simple function c() was well formulated.

**Weaknesses:**

- The model setup is slightly weak. For example, has consideration been given to stochastic c(·)? How easy is it to extend to unbounded domain beyond [0,1]^d?
- The definition of r_t says that when \hat{x}_t is not observable, r_t is f(x^*) - min_{x in X} f(x). This is just 0. Shouldn't regret be based on an output of the algorithm? If non-observable, the algorithm can simply output an element from observed data that it is sure is observable. Why are you defining regret to be 0 in such a case?
- There could be improvements in paper clarity. For example:

    > Please could you clean up the notations for kernel function k? Since it plays a central role in the theorems, it would be useful to setup the relationship between k(·) and the optimization function f(·) a little more cleanly.

- The authors claim that "Then, we provide the first regret upper bound of the GP optimization problem (1)..." But from my understanding, this well cited paper from 2009 provides an upperbound for GP-UCB (see Theorem 1)? https://arxiv.org/pdf/0912.3995.pdf


**Questions:**

- What is scaling of regret with dimension? Though your experiments indicate poor scaling with d, your regret expression doesn't explicitly show the dependence?
- \hat{T}^* = \theta_0/\eta)^{1/alpha} + 1, and regret is less thant B(\hat{T}^* -1). Why add and subtract 1? What is the order dependence on T?
- Typically people speak about cumulative, or simple regret. You have a theorem for regret r_t at time t. Can your theorem be written as a cumulative regret guarantee?
- What are your assumptions on C() that helps you to ignore the neighbourhood of a failure point? For example, adversarially, I could give c() to be a delta function at an arbitary number of points whose neighbourhood contains the optimal function. There are strong assumptions on f(), but the assumptions on c() were not clear to me.

**Limitations:**

- Seems like a minor improvement over non-failure aware GP based optimization Algorithms.
- If you disregard failures in GP-UCB, does it not converge? If so, why is the regret on that algorithm in your experiments never below 1 (what is the function c you have considered)?
- Seems like scaling with dimension d is quite poor? Does random restart help in such cases?

---

> ### Author Rebuttal · Authors · 2023-08-09
>
> We thank the reviewer for the comments.
> However, we believe that there are some misunderstandings in the "Weaknesses" section that the reviewer pointed out.
>
> > The definition of r_t says that when \hat{x}t is not observable, r_t is f(x^*) - min{x in X} f(x). This is just 0.
>
> Since our setup considers the maximization problem as in Eq. (1), $r_t$ does not become $0$ if $\hat{x}_t$ is not observable.
> Our definition of regret is based on the worst-case value of $f$ if $\hat{x}_t$ is not observable.
> It should be noted that this style of performance metric is common in constrained GP-optimization fields (see, e.g., [1]).
>
> [1]Hernández-Lobato et al., "A general framework for constrained Bayesian optimization using information-based search.", 2016
>
> > But from my understanding, this well cited paper from 2009 provides an upperbound for GP-UCB (see Theorem 1)?
>
> It should be noted that Theorem 1 of Srinivas et al. is not applicable to our problem setup.
> This is because, in our scenario, the standard GP-UCB, which disregards failures, leads to the posterior of the Gaussian process not being updated.
> Thus, the standard GP-UCB can be trapped at the same failure point.
> Our contribution lies in presenting the first algorithm for which an upper bound on regret is derived in the presence of unknown failure regions.
>
> > Has consideration been given to stochastic c(·)?
>
> While our paper does not consider the stochastic failure function $c$, it should be noted that practical applications where failures occur deterministically are not limited (e.g., the applications are discussed in [2]).
> Furthermore, algorithms suited for such scenarios differ from those designed for stochastic failures.
> (The distinctions between deterministic and stochastic failures are also examined in [2].)
>
> Given the fundamental dissimilarity between settings involving stochastic and deterministic failures, we believe that their theoretical analyses should be distinct and separate.
> For instance, even the definition of regret for a stochastic failure problem is not straightforward, underscoring the difference between these contexts.
>
> [2]Bachoc et al., "Gaussian process optimization with failures: classification and convergence proof.", 2020
>
>
>
> > How easy is it to extend to unbounded domain beyond [0,1]^d?
>
>
> From a theoretical standpoint, our algorithm can be extended to any compact convex input domain, as described in a footnote on page 4. (Unbounded input domain is not generally considered even in standard GP-optimization field.)
> The upper bound on regret for this extended version of the algorithm has already been provided in Appendix A.2 (Theorem A.6).
> It should be noted that the fundamental contribution of our paper, which is to build the first regret upper bound within the context of a failure-aware setup, is not limited by this aspect.
>
>
> From a practical standpoint, our algorithm requires to solve the set-covering problem over the input domain (as detailed in Lines 178-185).
> However, set-covering problem is NP-complete in general.
> Therefore, we provide a heuristic way (discussed in Lines 250-262) to avoid this issue for the input domain of bounded hypercube, which is one of the common settings in the GP-optimization field.
> Another common setting is a discrete input domain, in which the set-covering problem is easily solved. Thus, our algorithm is also applicable to the discrete input domain.
> The practical extension for the more general continuous input domain is an important research direction. We will carefully add the discussions in a revised version of our paper.
>
> We further give the answers to the reviewer's questions below.
>
> > What is scaling of regret with dimension? Though your experiments indicate poor scaling with d, your regret expression doesn't explicitly show the dependence?
>
> The dependence on $d$ in the regret upper bound is characterized by the maximum information gain $\gamma_{T; S_c}$ and $\tilde{T}_s$. The rate of the former is shown in Lines 112-114. The latter's definition is provided explicitly in Theorem 4.2.
>
> > \hat{T}^* = (\theta_0/\eta)^{1/alpha} + 1, and regret is less thant B(\hat{T}^* -1). Why add and subtract 1? What is the order dependence on T?
>
> Adding one is needed to make $\tilde{T}^*$ as the upper bound of the step, which search space of our algorithm contains the optimal solution (L226-227).
> We believe the current definition of $\tilde{T}^*$ will allow us to give an intuitive explanation that is helpful to readers to interpret our regret upper bound.
> Furthermore, it should be noted that $\tilde{T}^*$ is not dependent on $T$, but is the constant which only depends on failure function $c$ and the initial parameter of the algorithm.
>
>
> > Can your theorem be written as a cumulative regret guarantee?
>
> As described in a footnote of page 3, the cumulative regret guarantees of our algorithms are provided in Appendix C (Theorem C.1 and C.2)
>
>
> > What are your assumptions on C() that helps you to ignore the neighbourhood of a failure point?
> For example, ...
>
> Assumption 2.2 eliminates the case where the neighborhood of a failure point contains the optimal solution at all times.
> If $c$ is a function whose value is 1 (failure) at points that are arbitrarily close to the optimal solution, neighborhoods of their failure points can contain optimal solution at all times, i.e., Assumption 2.2 is violated.
> As concisely described in Lines 242-246 of Sec.4, we also discuss the possibility of loosening Assumption 2.2 in Appendix B.
>
>
>
> > If you disregard failures in GP-UCB, does it not converge?
>
> Standard GP-UCB in our setup does not converge.
> This is because the posterior of $f$ will not be updated when an observation is failed, and the algorithm could be trapped in the same failure point (Lines 329-330).
>
> We believe that most of the reviewer's concerns described in the "Weakness" section came from misunderstandings. We will carefully revise our paper to improve clarity.

---

> > ### Comment · Reviewer_Mr4k · 2023-08-14
> > **Raising Score**
> >
> > Thank you for your carefully made comments. I have increased confidence in the paper and it's soundness now. However my concerns on minor extension to GP-UCB still remain.
> >
> > I also have concerns over the dependence of regret with dimension, which is not clear to me. Can the dependence through the max-information-gain parameter be made more explicit? Is there some simple upper bound (even if loose) as to how regret scales with d?

---

> > > ### Author Response · Authors · 2023-08-15
> > >
> > > We thank the reviewer for the response.
> > >
> > > > Can the dependence through the max-information-gain parameter be made more explicit?
> > > Is there some simple upper bound (even if loose) as to how regret scales with d?
> > >
> > > The growing rate of maximum information gain $\gamma_{T;S_c}$ is dependent on an underlying kernel.
> > > For example, $\gamma_{T;S_c} = \mathcal{O}((\ln T)^{(d+1)})$ for Gaussian (squared exponential) kernel (Lines 235-238).
> > > Another common example is Matern kernel, whose $\gamma_{T;S_c}$ is $\mathcal{O}(T^{2\nu/(2\nu + d)} \ln (2\nu/(2\nu + d)))$
> > > with a smoothness parameter $\nu$ [1].
> > > It should be noted that these upper bounds of the maximum information gain are not specific to our setup but previous GP-optimization works have the same dependence of $d$ in a regret upper bound.
> > >
> > > As far as we know, no earlier work considers a simple (kernel-independent) upper bound of the maximum information gain since it might lead to disregarding the dependence of the kernel on regret.
> > >
> > > [1]Vakili et al., "On information gain and regret bounds in Gaussian process bandits.", 2021

---

### Official Review · Reviewer_GLK5 · 2023-07-24

**Soundness:** 3 good
**Presentation:** 3 good
**Contribution:** 3 good
**Rating:** 6
**Confidence:** 5

**Summary:**

This paper investigates GPBO under a robust setting, where the failure of observations is taken into account in the overall budget.  An adaptive algorithm based on GP-UCB has been proposed to address this problem, and both theoretical and practical behavior have been analyzed and verified.

**Strengths:**

Overall, this is a well-written paper with comprehensive analysis of the proposed algorithm. The empirical studies also demonstrate that the proposed algorithm is effective for at least low-dimensional problems.

**Weaknesses:**

The empirical behaviour of the proposed algorithm is only verified for dimensions less than 3, and it would be nicer to see more practical settings.  In such cases, a vanilla GP-UCB might work even better than a specially designed algorithm.  Thus, It might be interesting to explore the trade-offs in practice between the two approaches.

Minor:
line 154 "computes"

**Questions:**

To my knowledge, greedy elimination algorithms based on maximizing posterior variance usually achieve better theoratical convergence rates than UCB algorithms when it comes to robutsness considerations.  Do you have any idea that in the failure setting, a similar algorithm, say by replacing (5) with $x_t = \arg \max \sigma_t(x)$, would work better or worse?

---

> ### Author Rebuttal · Authors · 2023-08-09
>
> We thank the reviewer for the comments.
>
> > To my knowledge, greedy elimination algorithms based on maximizing posterior variance usually achieve better theoratical convergence rates than UCB algorithms when it comes to robutsness considerations.
> Do you have any idea that in the failure setting, a similar algorithm, say by replacing (5) with, would work better or worse?
>
> Please review the discussion in Appendix D.
> In summary, obstacles arise when considering elimination-based or maximum posterior variance-based algorithms in our setup, primarily due to unknown failure regions.
> Exploring a more theoretically well-behaved algorithm is an essential avenue for future research, and we leave this as a topic for future work.

---

### Decision · Program_Chairs · 2023-09-21

**Decision:**

Accept (poster)

**Comment:**

This paper tackles the problem of black-box optimization with observation failures in certain regions of the objective function. It proposes an extension of GP-UCB to this setting and provides a theoretical analysis that culminates in regret bounds.

There is consensus among the reviewers that the setting of BO with observation failures is important and that this paper provides a nice solution for it. The authors provide a comprehensive theoretical analysis and empirical evaluation. One reviewer liked in particular the experiment on quasicrystals. Several reviewers also raised questions (e.g. why the radius of the balls are cut in half or the definition of regret in rounds with observation failures) and concerns regarding clarity. The authors' responses could clarify those questions and some misunderstandings so that there is consensus that the paper is overall well presented (after some modifications).

There is a generally positive assessment of the paper by all reviewers after the discussion with the authors, but some general weaknesses of this work persist. Although the analysis seems well executed, it largely follows existing techniques. The algorithm is perhaps also a somewhat expected extension of GP-UCB. Further, the empirical evaluation is limited to 3-dimensional problems. Although high-dimensional problems are still extremely challenging for BO, 3 dimensions are still smaller than what is usually considered in BO benchmarks (6+ dimensions). This does leave some questions open regarding the applicability of this work.

Overall, this leaves the paper in the borderline regime. Since all reviewers are leaning positively, the paper is recommended to be accepted.